# EQUIVARIANT ENERGY-GUIDED SDE FOR INVERSE MOLECULAR DESIGN

**Fan Bao**[1] *, **Min Zhao**[1] *, **Zhongkai Hao**[1], **Peiyao Li**[1], **Chongxuan Li**[2 3] †, **Jun Zhu**[1] †
[1]Dept. of Comp. Sci. & Tech., Institute for AI, Tsinghua-Huawei Joint Center for AI
BNRist Center, State Key Lab for Intell. Tech. & Sys., Tsinghua University, Beijing, China
[2]Gaoling School of Artificial Intelligence, Renmin University of China, Beijing, China
[3]Beijing Key Laboratory of Big Data Management and Analysis Methods , Beijing, China
`bf19@mails.tsinghua.edu.cn, gracezhao1997@gmail.com,`
`{hzj21, lipy19}@mails.tsinghua.edu.cn,`
`chongxuanli@ruc.edu.cn, dcszj@tsinghua.edu.cn`

## ABSTRACT

Inverse molecular design is critical in material science and drug discovery, where the generated molecules should satisfy certain desirable properties. In this paper, we propose *equivariant energy-guided stochastic differential equations* (EEGSDE), a flexible framework for controllable 3D molecule generation under the guidance of an energy function in diffusion models. Formally, we show that EEGSDE naturally exploits the geometric symmetry in 3D molecular conformation, as long as the energy function is invariant to orthogonal transformations. Empirically, under the guidance of designed energy functions, EEGSDE significantly improves the baseline on QM9, in inverse molecular design targeted to quantum properties and molecular structures. Furthermore, EEGSDE is able to generate molecules with multiple target properties by combining the corresponding energy functions linearly.

## 1 INTRODUCTION

The discovery of new molecules with desired properties is critical in many fields, such as the drug and material design (Hajduk & Greer, 2007; Mandal et al., 2009; Kang et al., 2006; Pyzer-Knapp et al., 2015). However, brute-force search in the overwhelming molecular space is extremely challenging. Recently, inverse molecular design (Zunger, 2018) provides an efficient way to explore the molecular space, which directly predicts promising molecules that exhibit desired properties.

A natural way of inverse molecular design is to train a conditional generative model (Sanchez-Lengeling & Aspuru-Guzik, 2018). Formally, it learns a distribution of molecules conditioned on certain properties from data, and new molecules are predicted by sampling from the distribution with the condition set to desired properties. Among them, *equivariant diffusion models* (EDM) (Hoogeboom et al., 2022) leverage the current state-of-art diffusion models (Ho et al., 2020), which involves a forward process to perturb data and a reverse process to generate 3D molecules conditionally or unconditionally. While EDM generates stable and valid 3D molecules, we argue that a single conditional generative model is insufficient for generating accurate molecules that exhibit desired properties (see Table 1 and Table 3 for an empirical verification).

In this work, we propose *equivariant energy-guided stochastic differential equations* (EEGSDE), a flexible framework for controllable 3D molecule generation under the guidance of an energy function in diffusion models. EEGSDE formalizes the generation process as an equivariant stochastic differential equation, and plugs in energy functions to improve the controllability of generation. Formally, we show that EEGSDE naturally exploits the geometric symmetry in 3D molecular conformation, as long as the energy function is invariant to orthogonal transformations.

We apply EEGSDE to various applications by carefully designing task-specific energy functions. When targeted to quantum properties, EEGSDE is able to generate more accurate molecules than

---
*Equal contribution.    †Correspondence to: C. Li and J. Zhu.

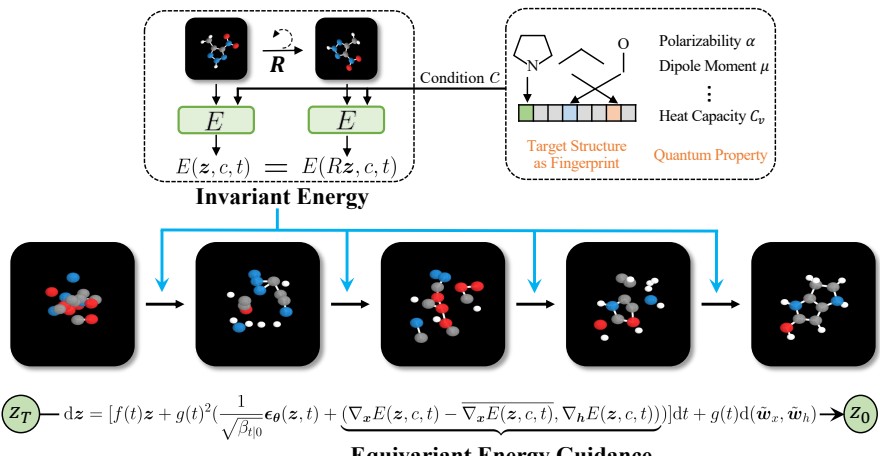

Figure 1: Overview of our EEGSDE. EEGSDE iteratively generates molecules with desired properties (represented by the condition $c$) by adopting the guidance of energy functions in each step. As the energy function is invariant to rotational transformation $\boldsymbol{R}$, its gradient (i.e., the energy guidance) is equivariant to $\boldsymbol{R}$, and therefore the distribution of generated samples is invariant to $\boldsymbol{R}$.

EDM, e.g., reducing the mean absolute error by more than 30% on the dipole moment property. When targeted to specific molecular structures, EEGSDE better capture the structure information in molecules than EDM, e.g, improving the similarity to target structures by more than 10%. Furthermore, EEGSDE is able to generate molecules targeted to multiple properties by combining the corresponding energy functions linearly. These demonstrate that our EEGSDE enables a flexible and controllable generation of molecules, providing a smart way to explore the chemical space.

## 2 RELATED WORK

**Diffusion models** are initially proposed by Sohl-Dickstein et al. (2015). Recently, they are better understood in theory by connecting it to score matching and stochastic differential equations (SDE) (Ho et al., 2020; Song et al., 2020). After that, diffusion models have shown strong empirical performance in many applications Dhariwal & Nichol (2021); Ramesh et al. (2022); Chen et al. (2020); Kong et al. (2020). There are also variants proposed to improve or accelerate diffusion models (Nichol & Dhariwal, 2021; Vahdat et al., 2021; Dockhorn et al., 2021; Bao et al., 2022b;a; Salimans & Ho, 2022; Lu et al., 2022).

**Guidance** is a technique to control the generation process of diffusion models. Initially, Song et al. (2020); Dhariwal & Nichol (2021) use classifier guidance to generate samples belonging to a class. Then, the guidance is extended to CLIP (Radford et al., 2021) for text to image generation, and semantic-aware energy (Zhao et al., 2022) for image-to-image translation. Prior guidance methods focus on image data, and are nontrivial to apply to molecules, since they do not consider the geometric symmetry. In contrast, our work proposes a general guidance framework for 3D molecules, where an invariant energy function is employed to leverage the geometric symmetry of molecules.

**Molecule generation.** Several works attempt to model molecules as 3D objects via deep generative models Nesterov et al. (2020); Gebauer et al. (2019); Satorras et al. (2021a); Hoffmann & Noé (2019); Hoogeboom et al. (2022). Among them, the most relevant one is the equivariant diffusion model (EDM) (Hoogeboom et al., 2022), which generates molecules in an iterative denoising manner. Benefiting from recent advances of diffusion models, EDM is stable to train and is able to generate high quality molecules. We provide a formal description of EDM in Section 3. Some other methods generate simplified representations of molecules, such as 1D SMILES strings (Weininger, 1988) and 2D graphs of molecules. These include variational autoencoders (Kusner et al., 2017; Dai et al., 2018; Jin et al., 2018; Simonovsky & Komodakis, 2018; Liu et al., 2018), normalizing flows (Madhawa et al., 2019; Zang & Wang, 2020; Luo et al., 2021), generative adversarial networks (Bian et al., 2019; Assouel et al., 2018), and autoregressive models (Popova et al., 2019; Flam-Shepherd et al., 2021). There are also methods on generating torsion angles in molecules. For

instance, Torsional Diffusion (Jing et al., 2022) employs the SDE formulation of diffusion models to model torsion angles in a given 2D molecular graph for the conformation generation task.

**Inverse molecular design.** Generative models have been applied to inverse molecular design. For example, conditional autoregressive models (Gebauer et al., 2022) and EDM (Hoogeboom et al., 2022) directly generate 3D molecules with desired quantum properties. Gebauer et al. (2019) also finetune pretrained generative models on a biased subset to generate 3D molecules with small HOMO-LUMO gaps. In contrast to these conditional generative models, our work further proposes a guidance method, a flexible way to control the generation process of molecules. Some other methods apply optimization methods to search molecules with desired properties, such as reinforcement learning (Zhou et al., 2019; You et al., 2018) and genetic algorithms (Jensen, 2019; Nigam et al., 2019). These optimization methods generally consider the 1D SMILES strings or 2D graphs of molecules, and the 3D information is not provided.

## 3 BACKGROUND

**3D representation of molecules.** Suppose a molecule has $M$ atoms and let $\boldsymbol{x}^i \in \mathbb{R}^n$ ($n = 3$ in general) be the coordinate of the $i$th atom. The collection of coordinates $\boldsymbol{x} = (\boldsymbol{x}^1, \ldots, \boldsymbol{x}^M) \in \mathbb{R}^{Mn}$ determines the *conformation* of the molecule. In addition to the coordinate, each atom is also associated with an atom feature, e.g., the atom type. We use $\boldsymbol{h}^i \in \mathbb{R}^d$ to represent the atom feature of the $i$th atom, and use $\boldsymbol{h} = (\boldsymbol{h}^1, \ldots, \boldsymbol{h}^M) \in \mathbb{R}^{Md}$ to represent the collection of atom features in a molecule. We use a tuple $\boldsymbol{z} = (\boldsymbol{x}, \boldsymbol{h})$ to represent a molecule, which contains both the 3D geometry information and the atom feature information.

**Equivariance and invariance.** Suppose $\boldsymbol{R}$ is a transformation. A distribution $p(\boldsymbol{x}, \boldsymbol{h})$ is said to be *invariant* to $\boldsymbol{R}$, if $p(\boldsymbol{x}, \boldsymbol{h}) = p(\boldsymbol{R}\boldsymbol{x}, \boldsymbol{h})$ holds for all $\boldsymbol{x}$ and $\boldsymbol{h}$. Here $\boldsymbol{R}\boldsymbol{x} = (\boldsymbol{R}\boldsymbol{x}^1, \ldots, \boldsymbol{R}\boldsymbol{x}^M)$ is applied to each coordinate. A function $(\boldsymbol{a}^x, \boldsymbol{a}^h) = \boldsymbol{f}(\boldsymbol{x}, \boldsymbol{h})$ that have two components $\boldsymbol{a}^x, \boldsymbol{a}^h$ in its output is said to be *equivariant* to $\boldsymbol{R}$, if $\boldsymbol{f}(\boldsymbol{R}\boldsymbol{x}, \boldsymbol{h}) = (\boldsymbol{R}\boldsymbol{a}^x, \boldsymbol{a}^h)$ holds for all $\boldsymbol{x}$ and $\boldsymbol{h}$. A function $\boldsymbol{f}(\boldsymbol{x}, \boldsymbol{h})$ is said to be *invariant* to $\boldsymbol{R}$, if $\boldsymbol{f}(\boldsymbol{R}\boldsymbol{x}, \boldsymbol{h}) = \boldsymbol{f}(\boldsymbol{x}, \boldsymbol{h})$ holds for all $\boldsymbol{x}$ and $\boldsymbol{h}$.

**Zero CoM subspace.** It has been shown that the invariance to translational and rotational transformations is an important factor for the success of 3D molecule modeling (Köhler et al., 2020; Xu et al., 2022). However, the translational invariance is impossible for a distribution in the full space $\mathbb{R}^{Mn}$ (Satorras et al., 2021a). Nevertheless, we can view two collections of coordinates $\boldsymbol{x}$ and $\boldsymbol{y}$ as equivalent if $\boldsymbol{x}$ can be translated from $\boldsymbol{y}$, since the translation doesn't change the identity of a molecule. Such an equivalence relation partitions the whole space $\mathbb{R}^{Mn}$ into disjoint equivalence classes. Indeed, all elements in the same equivalence classes represent the same conformation, and we can use the element with zero center of mass (CoM), i.e., $\frac{1}{M}\sum_{i=1}^M \boldsymbol{x}^i = \boldsymbol{0}$, as the specific representation. These elements collectively form the *zero CoM linear subspace* $X$ (Xu et al., 2022; Hoogeboom et al., 2022), and the rest of the paper always uses elements in $X$ to represent conformations.

**Equivariant graph neural network.** Satorras et al. (2021b) propose *equivariant graph neural networks* (EGNNs), which incorporate the equivariance inductive bias into neural networks. Specifically, $(\boldsymbol{a}^x, \boldsymbol{a}^h) = \text{EGNN}(\boldsymbol{x}, \boldsymbol{h})$ is a composition of $L$ *equivariant convolutional layers*. The $l$-th layer takes the tuple $(\boldsymbol{x}_l, \boldsymbol{h}_l)$ as the input and outputs an updated version $(\boldsymbol{x}_{l+1}, \boldsymbol{h}_{l+1})$, as follows:

$$\boldsymbol{m}^{ij} = \Phi_m(\boldsymbol{h}_l^i, \boldsymbol{h}_l^j, \|\boldsymbol{x}_l^i - \boldsymbol{x}_l^j\|_2^2, e^{ij}; \boldsymbol{\theta}_m), \ w^{ij} = \Phi_w(\boldsymbol{m}^{ij}; \boldsymbol{\theta}_w), \ \boldsymbol{h}_{l+1}^i = \Phi_h(\boldsymbol{h}_l^i, \sum_{j\neq i} w^{ij}\boldsymbol{m}^{ij}; \boldsymbol{\theta}_h),$$

$$\boldsymbol{x}_{l+1}^i = \boldsymbol{x}_l^i + \sum_{j\neq i} \frac{\boldsymbol{x}_l^i - \boldsymbol{x}_l^j}{\|\boldsymbol{x}_l^i - \boldsymbol{x}_l^j\|_2 + 1}\Phi_x(\boldsymbol{h}_l^i, \boldsymbol{h}_l^j, \|\boldsymbol{x}_l^i - \boldsymbol{x}_l^j\|_2^2, e^{ij}; \boldsymbol{\theta}_x),$$

where $\Phi_m, \Phi_w, \Phi_h, \Phi_x$ are parameterized by fully connected neural networks with parameters $\boldsymbol{\theta}_m, \boldsymbol{\theta}_w, \boldsymbol{\theta}_h, \boldsymbol{\theta}_x$ respectively, and $e^{ij}$ are optional feature attributes. We can verify that these layers are equivariant to orthogonal transformations, which include rotational transformations as special cases. As their composition, the EGNN is also equivariant to orthogonal transformations. Furthermore, let $\text{EGNN}^h(\boldsymbol{x}, \boldsymbol{h}) = \boldsymbol{a}^h$, i.e., the second component in the output of the EGNN. Then $\text{EGNN}^h(\boldsymbol{x}, \boldsymbol{h})$ is invariant to orthogonal transformations.

**Equivariant diffusion models (EDM)** (Hoogeboom et al., 2022) are a variant of diffusion models for molecule data. EDMs gradually inject noise to the molecule $\boldsymbol{z} = (\boldsymbol{x}, \boldsymbol{h})$ via a forward process

$$q(\boldsymbol{z}_{1:N}|\boldsymbol{z}_0) = \prod_{n=1}^{N} q(\boldsymbol{z}_n|\boldsymbol{z}_{n-1}), \ q(\boldsymbol{z}_n|\boldsymbol{z}_{n-1}) = \mathcal{N}_X(\boldsymbol{x}_n|\sqrt{\alpha_n}\boldsymbol{x}_{n-1}, \beta_n)\mathcal{N}(\boldsymbol{h}_n|\sqrt{\alpha_n}\boldsymbol{h}_{n-1}, \beta_n), \quad (1)$$

where $\alpha_n$ and $\beta_n$ represent the noise schedule and satisfy $\alpha_n + \beta_n = 1$, and $\mathcal{N}_X$ represent the Gaussian distribution in the zero CoM subspace $X$ (see its formal definition in Appendix A.2). Let $\overline{\alpha}_n = \alpha_1\alpha_2\cdots\alpha_n, \overline{\beta}_n = 1 - \overline{\alpha}_n$ and $\tilde{\beta}_n = \beta_n\overline{\beta}_{n-1}/\overline{\beta}_n$. To generate samples, the forward process is reversed using a Markov chain:

$$p(\boldsymbol{z}_{0:N}) = p(\boldsymbol{z}_N)\prod_{n=1}^{N} p(\boldsymbol{z}_{n-1}|\boldsymbol{z}_n), p(\boldsymbol{z}_{n-1}|\boldsymbol{z}_n) = \mathcal{N}_X(\boldsymbol{x}_{n-1}|\boldsymbol{\mu}_n^x(\boldsymbol{z}_n), \tilde{\beta}_n)\mathcal{N}(\boldsymbol{h}_{n-1}|\boldsymbol{\mu}_n^h(\boldsymbol{z}_n), \tilde{\beta}_n). \quad (2)$$

Here $p(\boldsymbol{z}_N) = \mathcal{N}_X(\boldsymbol{x}_N|\boldsymbol{0}, 1)\mathcal{N}(\boldsymbol{h}_N|\boldsymbol{0}, 1)$. The mean $\boldsymbol{\mu}_n(\boldsymbol{z}_n) = (\boldsymbol{\mu}_n^x(\boldsymbol{z}_n), \boldsymbol{\mu}_n^h(\boldsymbol{z}_n))$ is parameterized by a noise prediction network $\boldsymbol{\epsilon_\theta}(\boldsymbol{z}_n, n)$, and is trained using a MSE loss, as follows:

$$\boldsymbol{\mu}_n(\boldsymbol{z}_n) = \frac{1}{\sqrt{\alpha_n}}(\boldsymbol{z}_n - \frac{\beta_n}{\sqrt{\overline{\beta}_n}}\boldsymbol{\epsilon_\theta}(\boldsymbol{z}_n, n)), \quad \min_{\boldsymbol{\theta}} \mathbb{E}_n\mathbb{E}_{q(\boldsymbol{z}_0,\boldsymbol{z}_n)}w(t)\|\boldsymbol{\epsilon_\theta}(\boldsymbol{z}_n, n) - \boldsymbol{\epsilon}_n\|^2,$$

where $\boldsymbol{\epsilon}_n = \frac{\boldsymbol{z}_n - \sqrt{\overline{\alpha}_n}\boldsymbol{z}_0}{\sqrt{\overline{\beta}_n}}$ is the standard Gaussian noise injected to $\boldsymbol{z}_0$ and $w(t)$ is the weight term.

Hoogeboom et al. (2022) show that the distribution of generated samples $p(\boldsymbol{z}_0)$ is invariant to rotational transformations if the noise prediction network is equivariant to orthogonal transformations. In Section 4.2, we extend this proposition to the SDE formulation of molecular diffusion modelling.

Hoogeboom et al. (2022) also present a conditional version of EDM for inverse molecular design by adding an extra input of the condition $c$ to the noise prediction network as $\boldsymbol{\epsilon_\theta}(\boldsymbol{z}_n, c, n)$.

## 4 EQUIVARIANT ENERGY-GUIDED SDE

In this part, we introduce our equivariant energy-guided SDE (EEGSDE), as illustrated in Figure 1. EEGSDE is based on the SDE formulation of molecular diffusion modeling, which is described in Section 4.1 and Section 4.2. Then, we formally present our EEGSDE that incorporates an energy function to guide the molecular generation in Section 4.3. We provide derivations in Appendix A.

### 4.1 SDE IN THE PRODUCT SPACE

Recall that a molecule is represented as a tuple $\boldsymbol{z} = (\boldsymbol{x}, \boldsymbol{h})$, where $\boldsymbol{x} = (\boldsymbol{x}^1, \ldots, \boldsymbol{x}^M) \in X$ represents the conformation and $\boldsymbol{h} = (\boldsymbol{h}^1, \ldots, \boldsymbol{h}^M) \in \mathbb{R}^{Md}$ represents atom features. Here $X = \{\boldsymbol{x} \in \mathbb{R}^{Mn} : \frac{1}{M}\sum_{i=1}^{M}\boldsymbol{x}^i = \boldsymbol{0}\}$ is the zero CoM subspace mentioned in Section 3, and $d$ is the feature dimension. We first introduce a continuous-time diffusion process $\{\boldsymbol{z}_t\}_{0 \leq t \leq T}$ in the product space $X \times \mathbb{R}^{Md}$, which gradually adds noise to $\boldsymbol{x}$ and $\boldsymbol{h}$. This can be described by the forward SDE:

$$d\boldsymbol{z} = f(t)\boldsymbol{z}dt + g(t)d(\boldsymbol{w}_x, \boldsymbol{w}_h), \quad \boldsymbol{z}_0 \sim q(\boldsymbol{z}_0), \quad (3)$$

where $f(t)$ and $g(t)$ are two scalar functions, $\boldsymbol{w}_x$ and $\boldsymbol{w}_h$ are independent standard Wiener processes in $X$ and $\mathbb{R}^{Md}$ respectively, and the SDE starts from the data distribution $q(\boldsymbol{z}_0)$. Note that $\boldsymbol{w}_x$ can be constructed by subtracting the CoM of a standard Wiener process $\boldsymbol{w}$ in $\mathbb{R}^{Mn}$, i.e., $\boldsymbol{w}_x = \boldsymbol{w} - \overline{\boldsymbol{w}}$, where $\overline{\boldsymbol{w}} = \frac{1}{M}\sum_{i=1}^{M}\boldsymbol{w}^i$ is the CoM of $\boldsymbol{w} = (\boldsymbol{w}^1, \ldots, \boldsymbol{w}^M)$. It can be shown that the SDE has a linear Gaussian transition kernel $q(\boldsymbol{z}_t|\boldsymbol{z}_s) = q(\boldsymbol{x}_t|\boldsymbol{x}_s)q(\boldsymbol{h}_t|\boldsymbol{h}_s)$ from $\boldsymbol{x}_s$ to $\boldsymbol{x}_t$, where $0 \leq s < t \leq T$. Specifically, there exists two scalars $\alpha_{t|s}$ and $\beta_{t|s}$, s.t., $q(\boldsymbol{x}_t|\boldsymbol{x}_s) = \mathcal{N}_X(\boldsymbol{x}_t|\sqrt{\alpha_{t|s}}\boldsymbol{x}_s, \beta_{t|s})$ and $q(\boldsymbol{h}_t|\boldsymbol{h}_s) = \mathcal{N}(\boldsymbol{h}_t|\sqrt{\alpha_{t|s}}\boldsymbol{h}_s, \beta_{t|s})$. Here $\mathcal{N}_X$ denotes the Gaussian distribution in the subspace $X$, and see Appendix A.2 for its formal definition. Indeed, the forward process of EDM in Eq. (1) is a discretization of the forward SDE in Eq. (3).

To generate molecules, we reverse Eq. (3) from $T$ to 0. Such a time reversal forms another a SDE, which can be represented by both the *score function form* and the *noise prediction form*:

$$d\boldsymbol{z} = [f(t)\boldsymbol{z} - g(t)^2 \underbrace{(\nabla_{\boldsymbol{x}} \log q_t(\boldsymbol{z}) - \overline{\nabla_{\boldsymbol{x}} \log q_t(\boldsymbol{z})}, \nabla_{\boldsymbol{h}} \log q_t(\boldsymbol{z}))}_{\text{score function form}}]dt + g(t)d(\tilde{\boldsymbol{w}}_x, \tilde{\boldsymbol{w}}_h),$$

$$= [f(t)\boldsymbol{z} + \frac{g(t)^2}{\sqrt{\beta_{t|0}}} \underbrace{\mathbb{E}_{q(\boldsymbol{z}_0|\boldsymbol{z}_t)}\boldsymbol{\epsilon}_t}_{\text{noise prediction form}}]dt + g(t)d(\tilde{\boldsymbol{w}}_x, \tilde{\boldsymbol{w}}_h), \quad \boldsymbol{z}_T \sim q_T(\boldsymbol{z}_T). \quad (4)$$

Here $q_t(\boldsymbol{z})$ is the marginal distribution of $\boldsymbol{z}_t$, $\nabla_{\boldsymbol{x}} \log q_t(\boldsymbol{z})$ is the gradient of $\log q_t(\boldsymbol{z})$ w.r.t. $\boldsymbol{x}$[1], $\overline{\nabla_{\boldsymbol{x}} \log q_t(\boldsymbol{z})} = \frac{1}{M}\sum_{i=1}^{M} \nabla_{\boldsymbol{x}^i} \log q_t(\boldsymbol{z})$ is the CoM of $\nabla_{\boldsymbol{x}} \log q_t(\boldsymbol{z})$, $dt$ is the infinitesimal negative timestep, $\tilde{\boldsymbol{w}}_x$ and $\tilde{\boldsymbol{w}}_h$ are independent reverse-time standard Wiener processes in $X$ and $\mathbb{R}^{Md}$ respectively, and $\boldsymbol{\epsilon}_t = \frac{\boldsymbol{z}_t - \sqrt{\alpha_{t|0}}\boldsymbol{z}_0}{\sqrt{\beta_{t|0}}}$ is the standard Gaussian noise injected to $\boldsymbol{z}_0$. Compared to the original SDE introduced by Song et al. (2020), our reverse SDE in Eq. (4) additionally subtracts the CoM of $\nabla_{\boldsymbol{x}} \log q_t(\boldsymbol{z})$. This ensures $\boldsymbol{x}_t$ always stays in the zero CoM subspace as time flows back.

To sample from the reverse SDE in Eq. (4), we use a noise prediction network $\boldsymbol{\epsilon}_{\boldsymbol{\theta}}(\boldsymbol{z}_t, t)$ to estimate $\mathbb{E}_{q(\boldsymbol{z}_0|\boldsymbol{z}_t)}\boldsymbol{\epsilon}_t$, through minimizing the MSE loss $\min_{\boldsymbol{\theta}} \mathbb{E}_t \mathbb{E}_{q(\boldsymbol{z}_0, \boldsymbol{z}_t)} w(t)\|\boldsymbol{\epsilon}_{\boldsymbol{\theta}}(\boldsymbol{z}_t, t) - \boldsymbol{\epsilon}_t\|^2$, where $t$ is uniformly sampled from $[0, T]$, and $w(t)$ controls the weight of the loss term at time $t$. Note that the noise $\boldsymbol{\epsilon}_t$ is in the product space $X \times \mathbb{R}^{Md}$, so we subtract the CoM of the predicted noise of $\boldsymbol{x}_t$ to ensure $\boldsymbol{\epsilon}_{\boldsymbol{\theta}}(\boldsymbol{z}_t, t)$ is also in the product space.

Substituting $\boldsymbol{\epsilon}_{\boldsymbol{\theta}}(\boldsymbol{z}_t, t)$ into Eq. (4), we get an approximate reverse-time SDE parameterized by $\boldsymbol{\theta}$:

$$d\boldsymbol{z} = [f(t)\boldsymbol{z} + \frac{g(t)^2}{\sqrt{\beta_{t|0}}}\boldsymbol{\epsilon}_{\boldsymbol{\theta}}(\boldsymbol{z}, t)]dt + g(t)d(\tilde{\boldsymbol{w}}_x, \tilde{\boldsymbol{w}}_h), \quad \boldsymbol{z}_T \sim p_T(\boldsymbol{z}_T), \quad (5)$$

where $p_T(\boldsymbol{z}_T) = \mathcal{N}_X(\boldsymbol{x}_T|\boldsymbol{0}, 1)\mathcal{N}(\boldsymbol{h}_T|\boldsymbol{0}, 1)$ is a Gaussian prior in the product space that approximates $q_T(\boldsymbol{z}_T)$. We define $p_{\boldsymbol{\theta}}(\boldsymbol{z}_0)$ as the marginal distribution of Eq. (5) at time $t = 0$, which is the distribution of our generated samples. Similarly to the forward process, the reverse process of EDM in Eq. (2) is a discretization of the reverse SDE in Eq. (5).

## 4.2 EQUIVARIANT SDE

To leverage the geometric symmetry in 3D molecular conformation, $p_{\boldsymbol{\theta}}(\boldsymbol{z}_0)$ should be invariant to translational and rotational transformations. As mentioned in Section 3, the translational invariance of $p_{\boldsymbol{\theta}}(\boldsymbol{z}_0)$ is already satisfied by considering the zero CoM subspace. The rotational invariance can be satisfied if the noise prediction network is equivariant to orthogonal transformations, as summarized in the following theorem:

**Theorem 1.** *Let* $(\boldsymbol{\epsilon}_{\boldsymbol{\theta}}^x(\boldsymbol{z}_t, t), \boldsymbol{\epsilon}_{\boldsymbol{\theta}}^h(\boldsymbol{z}_t, t)) = \boldsymbol{\epsilon}_{\boldsymbol{\theta}}(\boldsymbol{z}_t, t)$, *where* $\boldsymbol{\epsilon}_{\boldsymbol{\theta}}^x(\boldsymbol{z}_t, t)$ *and* $\boldsymbol{\epsilon}_{\boldsymbol{\theta}}^h(\boldsymbol{z}_t, t)$ *are the predicted noise of* $\boldsymbol{x}_t$ *and* $\boldsymbol{h}_t$ *respectively. If for any orthogonal transformation* $\boldsymbol{R} \in \mathbb{R}^{n \times n}$, $\boldsymbol{\epsilon}_{\boldsymbol{\theta}}(\boldsymbol{z}_t, t)$ *is equivariant to* $\boldsymbol{R}$, *i.e.,* $\boldsymbol{\epsilon}_{\boldsymbol{\theta}}(\boldsymbol{R}\boldsymbol{x}_t, \boldsymbol{h}_t, t) = (\boldsymbol{R}\boldsymbol{\epsilon}_{\boldsymbol{\theta}}^x(\boldsymbol{x}_t, \boldsymbol{h}_t, t), \boldsymbol{\epsilon}_{\boldsymbol{\theta}}^h(\boldsymbol{x}_t, \boldsymbol{h}_t, t))$, *and* $p_T(\boldsymbol{z}_T)$ *is invariant to* $\boldsymbol{R}$, *i.e.,* $p_T(\boldsymbol{R}\boldsymbol{x}_T, \boldsymbol{h}_T) = p_T(\boldsymbol{x}_T, \boldsymbol{h}_T)$, *then* $p_{\boldsymbol{\theta}}(\boldsymbol{z}_0)$ *is invariant to any rotational transformation.*

As mentioned in Section 3, the EGNN satisfies the equivariance constraint, and we parameterize $\boldsymbol{\epsilon}_{\boldsymbol{\theta}}(\boldsymbol{z}_t, t)$ using an EGNN following Hoogeboom et al. (2022). See details in Appendix D.

## 4.3 EQUIVARIANT ENERGY-GUIDED SDE

Now we describe *equivariant energy-guided SDE* (EEGSDE), which guides the generated molecules of Eq. (5) towards desired properties $c$ by leveraging a time-dependent energy function $E(\boldsymbol{z}, c, t)$:

$$d\boldsymbol{z} = [f(t)\boldsymbol{z} + g(t)^2(\frac{1}{\sqrt{\beta_{t|0}}}\boldsymbol{\epsilon}_{\boldsymbol{\theta}}(\boldsymbol{z}, t)$$

$$+ \underbrace{(\nabla_{\boldsymbol{x}}E(\boldsymbol{z}, c, t) - \overline{\nabla_{\boldsymbol{x}}E(\boldsymbol{z}, c, t)}, \nabla_{\boldsymbol{h}}E(\boldsymbol{z}, c, t))}_{\text{energy gradient taken in the product space}})]dt + g(t)d(\tilde{\boldsymbol{w}}_x, \tilde{\boldsymbol{w}}_h), \, \boldsymbol{z}_T \sim p_T(\boldsymbol{z}_T), \, (6)$$

---

[1]While $q_t(\boldsymbol{z})$ is defined in $X \times \mathbb{R}^{Md}$, its domain can be extended to $\mathbb{R}^{Mn} \times \mathbb{R}^{Md}$ and the gradient is valid. See Remark 1 in Appendix A.2 for details.

which defines a distribution $p_{\boldsymbol{\theta}}(\boldsymbol{z}_0|c)$ conditioned on the property $c$. Here the CoM $\overline{\nabla_{\boldsymbol{x}} E(\boldsymbol{z}, c, t)}$ of the gradient is subtracted to keep the SDE in the product space, which ensures the translational invariance of $p_{\boldsymbol{\theta}}(\boldsymbol{z}_0|c)$. Besides, the rotational invariance is satisfied by using energy invariant to orthogonal transformations, as summarized in the following theorem:

**Theorem 2.** *Suppose the assumptions in Theorem 1 hold and $E(\boldsymbol{z}, c, t)$ is invariant to any orthogonal transformation $\boldsymbol{R}$, i.e., $E(\boldsymbol{R}\boldsymbol{x}, \boldsymbol{h}, c, t) = E(\boldsymbol{x}, \boldsymbol{h}, c, t)$. Then $p_{\boldsymbol{\theta}}(\boldsymbol{z}_0|c)$ is invariant to any rotational transformation.*

Note that we can also use a conditional model $\boldsymbol{\epsilon_\theta}(\boldsymbol{z}, c, t)$ in Eq. (6). See Appendix C for details. To sample from $p_{\boldsymbol{\theta}}(\boldsymbol{z}_0|c)$, various solvers can be used for Eq. (6), such as the Euler-Maruyama method (Song et al., 2020) and the Analytic-DPM sampler (Bao et al., 2022b;a). We present the Euler-Maruyama method as an example in Algorithm 1 at Appendix B.

### 4.4 HOW TO DESIGN THE ENERGY FUNCTION

Our EEGSDE is a general framework, which can be applied to various applications by specifying different energy functions $E(\boldsymbol{z}, c, t)$. For example, we can design the energy function according to consistency between the molecule $\boldsymbol{z}$ and the property $c$, where a low energy represents a well consistency. As the generation process in Eq. (6) proceeds, the gradient of the energy function encourages generated molecules to have a low energy, and consequently a well consistency. Thus, we can expect the generated molecule $\boldsymbol{z}$ aligns well with the property $c$. In the rest of the paper, we specify the choice of energy functions, and show these energies improve controllable molecule generation targeted to quantum properties, molecular structures, and even a combination of them.

**Remark.** The term "energy" in this paper refers to a general notion in statistical machine learning, which is a scalar function that captures dependencies between input variables (LeCun et al., 2006). Thus, the "energy" in this paper can be set to a MSE loss when we want to capture how the molecule align with the property (as done in Section 5). Also, the "energy" in this paper does not exclude potential energy or free energy in chemistry, and they might be applicable when we want to generate molecules with small potential energy or free energy.

## 5 GENERATING MOLECULES WITH DESIRED QUANTUM PROPERTIES

Let $c \in \mathbb{R}$ be a certain quantum property. To generate molecules with the desired property, we set the energy function as the squared error between the predicted property and the desired property $E(\boldsymbol{z}_t, c, t) = s|g(\boldsymbol{z}_t, t) - c|^2$, where $g(\boldsymbol{z}_t, t)$ is a time-dependent property prediction model, and $s$ is the scaling factor controlling the strength of the guidance. Specifically, $g(\boldsymbol{z}_t, t)$ can be parameterized by equivariant models such as EGNN (Satorras et al., 2021b), SE3-Transformer (Fuchs et al., 2020) and DimeNet (Klicpera et al., 2020) to ensure the invariance of $E(\boldsymbol{z}_t, c, t)$, as long as they perform well in the task of property prediction. In this paper we consider EGNN. We provide details on parameterization and the training objective in Appendix E. We can also generate molecules targeted to multiple quantum properties by combining energy functions linearly (see details in Appendix F.1).

### 5.1 SETUP

We evaluate on QM9 (Ramakrishnan et al., 2014), which contains quantum properties and coordinates of $\sim$130k molecules with up to nine heavy atoms from (C, N, O, F). Following EDM, we split QM9 into training, validation and test sets, which include 100K, 18K and 13K samples respectively. The training set is further divided into two non-overlapping halves $D_a, D_b$ equally. The noise prediction network and the time-dependent property prediction model of the energy function are trained on $D_b$ separately. By default, EEGSDE uses a conditional noise prediction network $\boldsymbol{\epsilon_\theta}(\boldsymbol{z}, c, t)$ in Eq. (6), since we find it generate more accurate molecules than using an unconditional one (see Appendix G.1 for an ablation study). See more details in Appendix F.3.

**Evaluation metric:** Following EDM, we use the mean absolute error (MAE) to evaluate how generated molecules align with the desired property. Specifically, we train another property prediction model $\phi_p$ (Satorras et al., 2021b) on $D_a$, and the MAE is calculated as $\frac{1}{K}\sum_{i=1}^{K}|\phi_p(\boldsymbol{z}_i) - c_i|$, where $\boldsymbol{z}_i$ is a generated molecule, $c_i$ is its desired property. We generate $K$=10,000 samples for evaluation with $\phi_p$. For fairness, the property prediction model $\phi_p$ is different from $g(\boldsymbol{z}_t, t)$ used in the energy

Table 1: How generated molecules align with the target quantum property. The L-bound (Hoogeboom et al., 2022) represents the loss of $\phi_p$ on $D_b$ and can be viewed as a lower bound of the MAE metric. The conditional EDM results are reproduced, and are consistent with Hoogeboom et al. (2022) (see Appendix G.4). "#Atoms" uses public results from Hoogeboom et al. (2022).

| Method | MAE↓ | Method | MAE↓ | Method | MAE↓ |
|---|---|---|---|---|---|
| $C_v$ ($\frac{\text{cal}}{\text{mol}}$K) | | $\mu$ (D) | | $\alpha$ (Bohr$^3$) | |
| U-bound | 6.879±0.015 | U-bound | 1.613±0.003 | U-bound | 8.98±0.02 |
| #Atoms | 1.971 | #Atoms | 1.053 | #Atoms | 3.86 |
| Conditional EDM | 1.065±0.010 | Conditional EDM | 1.123±0.013 | Conditional EDM | 2.78±0.04 |
| EEGSDE ($s$=1) | 1.037±0.010 | EEGSDE ($s$=0.5) | 0.930±0.005 | EEGSDE ($s$=0.5) | 2.67±0.04 |
| EEGSDE ($s$=5) | 0.981±0.002 | EEGSDE ($s$=1) | 0.858±0.006 | EEGSDE ($s$=1) | 2.62±0.03 |
| EEGSDE ($s$=10) | **0.941**±0.005 | EEGSDE ($s$=2) | **0.777**±0.007 | EEGSDE ($s$=3) | **2.50**±0.02 |
| L-bound | 0.040 | L-bound | 0.043 | L-bound | 0.09 |
| $\Delta\varepsilon$ (meV) | | $\varepsilon_{\text{HOMO}}$ (meV) | | $\varepsilon_{\text{LUMO}}$ (meV) | |
| U-bound | 1464±4 | U-bound | 645±41 | U-bound | 1457±5 |
| #Atoms | 866 | #Atoms | 426 | #Atoms | 813 |
| Conditional EDM | 671±5 | Conditional EDM | 371±2 | Conditional EDM | 601±7 |
| EEGSDE ($s$=0.5) | 574±4 | EEGSDE ($s$=0.1) | 357±4 | EEGSDE ($s$=0.5) | 525±4 |
| EEGSDE ($s$=1) | 542±2 | EEGSDE ($s$=0.5) | 320±1 | EEGSDE ($s$=1) | 496±2 |
| EEGSDE ($s$=3) | **487**±3 | EEGSDE ($s$=1) | **302**±2 | EEGSDE ($s$=3) | **447**±6 |
| L-bound | 65 | L-bound | 39 | L-bound | 36 |

Table 2: The mean absolute error (MAE) computed by the Gaussian software instead of $\phi_p$.

| Method | MAE↓ | Method | MAE↓ | Method | MAE↓ | Method | MAE↓ | Method | MAE↓ |
|---|---|---|---|---|---|---|---|---|---|
| $\mu$ (D) | | $\alpha$ (Bohr$^3$) | | $\Delta\varepsilon$ (meV) | | $\varepsilon_{\text{HOMO}}$ (meV) | | $\varepsilon_{\text{LUMO}}$ (meV) | |
| Conditional EDM | 1.20 | Conditional EDM | 2.41 | Conditional EDM | 775 | Conditional EDM | 354 | Conditional EDM | 573 |
| EEGSDE ($s$=0.5) | 0.96 | EEGSDE ($s$=0.5) | 2.27 | EEGSDE ($s$=0.5) | 638 | EEGSDE ($s$=0.1) | 349 | EEGSDE ($s$=0.5) | 495 |
| EEGSDE ($s$=1) | 0.78 | EEGSDE ($s$=1) | 2.03 | EEGSDE ($s$=1) | 555 | EEGSDE ($s$=0.5) | 341 | EEGSDE ($s$=1) | 445 |
| EEGSDE ($s$=2) | **0.73** | EEGSDE ($s$=3) | **1.85** | EEGSDE ($s$=3) | **532** | EEGSDE ($s$=1) | **284** | EEGSDE ($s$=3) | **416** |

Table 3: How generated molecules align with multiple target quantum properties.

| Method | MAE1↓ | MAE2↓ |
|---|---|---|
| $C_v$ ($\frac{\text{cal}}{\text{mol}}$K), $\mu$ (D) | | |
| Conditional EDM | 1.079±0.007 | 1.156±0.011 |
| EEGSDE ($s_1$=10, $s_2$=1) | **0.981**±0.008 | **0.912**±0.006 |
| $\Delta\varepsilon$ (meV), $\mu$ (D) | | |
| Conditional EDM | 683±1 | 1.130±0.007 |
| EEGSDE ($s_1$=$s_2$=1) | **563**±3 | **0.866**±0.003 |
| $\alpha$ (Bohr$^3$), $\mu$ (D) | | |
| Conditional EDM | 2.76±0.01 | 1.158±0.002 |
| EEGSDE ($s_1$=$s_2$=1.5) | **2.61**±0.01 | **0.855**±0.007 |

Table 4: How generated molecules align with target structures.

| Method | Similarity↑ |
|---|---|
| QM9 | |
| cG-SchNet | 0.499±0.002 |
| Conditional EDM | 0.671±0.004 |
| EEGSDE ($s$=0.1) | 0.696±0.002 |
| EEGSDE ($s$=0.5) | 0.736±0.002 |
| EEGSDE ($s$=1.0) | **0.750**±0.003 |
| GEOM-Drug | |
| Conditional EDM | 0.165±0.001 |
| EEGSDE ($s$=0.5) | 0.185±0.001 |
| EEGSDE ($s$=1.0) | **0.193**±0.001 |

function, and they are trained on the two non-overlapping training subsets $D_a, D_b$ respectively. This ensures no information leak occurs when evaluating our EEGSDE. To further verify the effectiveness of EEGSDE, we also calculate the MAE without relying on a neural network. Specifically, we use the Gaussian software (which calculates properties according to theories of quantum chemistry without using neural networks) to calculate the properties of 100 generated molecules. For completeness, we also report the novelty, the atom stability and the molecule stability following Hoogeboom et al. (2022) in Appendix G.2, although they are not our main focus.

**Baseline:** The most direct baseline is conditional EDM, which only adopts a conditional noise prediction network. We also compare two additional baselines "U-bound" and "#Atoms" from Hoogeboom et al. (2022) (see Appendix F.2 for details).

## 5.2 RESULTS

Following Hoogeboom et al. (2022), we consider six quantum properties in QM9: polarizability $\alpha$, highest occupied molecular orbital energy $\varepsilon_{\text{HOMO}}$, lowest unoccupied molecular orbital energy $\varepsilon_{\text{LUMO}}$, HOMO-LUMO gap $\Delta\varepsilon$, dipole moment $\mu$ and heat capacity $C_v$. Firstly, we generate

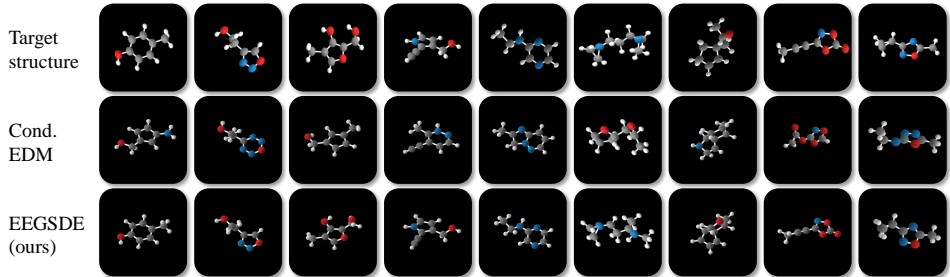

Figure 2: Generated molecules on QM9 targeted to specific structures (unseen during training). The molecular structures of EEGSDE align better with target structures then conditional EDM.

molecules targeted to one of these six properties. As shown in Table 1, with the energy guidance, our EEGSDE has a significantly better MAE than the conditional EDM on all properties. Remarkably, with a proper scaling factor $s$, the MAE of EEGSDE is reduced by more than 25% compared to conditional EDM on properties $\Delta\varepsilon$, $\varepsilon_{\text{LUMO}}$, and more than 30% on $\mu$. What's more, as shown in Table 2, our EEGSDE still has better MAE under the evaluation by the Gaussian software, which further verifies the effectiveness of our EEGSDE. We further generate molecules targeted to multiple quantum properties by combining energy functions linearly. As shown in Table 3, our EEGSDE still has a significantly better MAE than the conditional EDM.

**Conclusion:** These results suggest that molecules generated by our EEGSDE align better with the desired properties than molecules generated by the conditional EDM baseline (for both the single-property and multiple-properties cases). As a consequence, EEGSDE is able to explore the chemical space in a guided way to generate promising molecules for downstream applications such as the virtual screening, which may benefit drug and material discovery.

## 6 GENERATING MOLECULES WITH TARGET STRUCTURES

Following Gebauer et al. (2022), we use the molecular fingerprint to encode the structure information of a molecule. The molecular fingerprint $c = (c_1, \ldots, c_L)$ is a series of bits that capture the presence or absence of substructures in the molecule. Specifically, a substructure is mapped to a specific position $l$ in the bitmap, and the corresponding bit $c_l$ will be 1 if the substructure exists in the molecule and will be 0 otherwise. To generate molecules with a specific structure (encoded by the fingerprint $c$), we set the energy function as the squared error $E(\boldsymbol{z}_t, c, t) = s\|m(\boldsymbol{z}_t, t) - c\|^2$ between a time-dependent multi-label classifier $m(\boldsymbol{z}_t, t)$ and $c$. Here $s$ is the scaling factor, and $m(\boldsymbol{z}_t, t)$ is trained with binary cross entropy loss to predict the fingerprint as detailed in Appendix E.2. Note that the choice of the energy function is flexible and can be different to the training loss of $m(\boldsymbol{z}_t, t)$. In initial experiments, we also try binary cross entropy loss for the energy function, but we find it causes the generation process unstable. The multi-label classifier $m(\boldsymbol{z}_t, t)$ is parameterized in a similar way to the property prediction model $g(\boldsymbol{z}_t, t)$ in Section 5, and we present details in Appendix E.1.

### 6.1 SETUP

We evaluate on QM9 and GEOM-Drug. We train our method (including the noise prediction network and the multi-label classifier) on the whole training set. By default, we use a conditional noise prediction network $\boldsymbol{\epsilon_\theta}(\boldsymbol{z}, c, t)$ in Eq. (6) for a better performance. See more details in Appendix F.4.

**Evaluation metric:** To measure how the structure of a generated molecule aligns with the target one, we use the Tanimoto similarity (Gebauer et al., 2022), which captures similarity between structures by comparing their fingerprints.

**Baseline:** The most direct baseline is conditional EDM (Hoogeboom et al., 2022), which only adopts a conditional noise prediction network without the guidance of an energy model. We also consider cG-SchNet (Gebauer et al., 2022), which generates molecules in an autoregressive manner.

### 6.2 RESULTS

As shown in Table 4, EEGSDE significantly improves the similarity between target structures and generated structures compared to conditional EDM and cG-SchNet on QM9. Also note in a proper range, a larger scaling factor results in a better similarity, and EEGSDE with $s$=1 improves the sim-

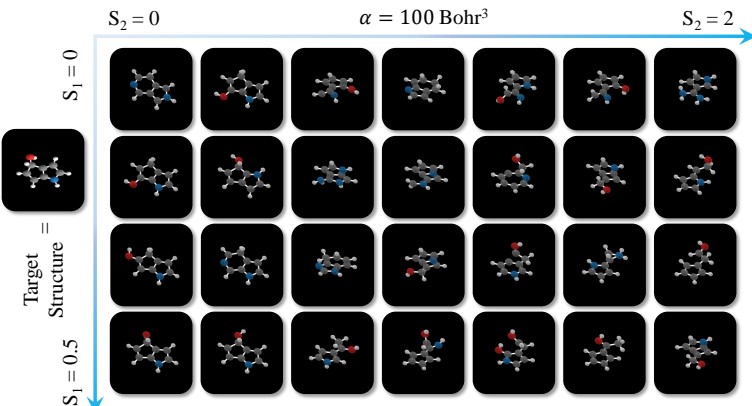

Figure 3: Generate molecules on QM9 targeted to both the quantum property $\alpha$ and the molecular structure. As the scaling factor $s_2$ grows, the substructure of generated molecule gradually change from the symmetric ring to a less isometrically shaped structure. Meanwhile the generated molecule aligns better with the target structure as the scaling factor $s_1$ grows.

ilarity by more than 10% compared to conditional EDM. In Figure 2, we plot generated molecules of conditional EDM and EEGSDE ($s$=1) targeted to specific structures, where our EEGSDE aligns better with them. We further visualize the effect of the scaling factor in Appendix G.3, where the generated structures align better as the scaling factor grows. These results demonstrate that our EEGSDE captures the structure information in molecules well.

We also perform experiments on the more challenging GEOM-Drug (Axelrod & Gomez-Bombarelli, 2022) dataset, and we train the conditional EDM baseline following the default setting of Hooge-boom et al. (2022). As shown in Table 4, we find the conditional EDM baseline has a similarity of 0.165, which is much lower than the value on QM9. We hypothesize this is because molecules in GEOM-Drug has much more atoms than QM9 with a more complex structure, and the default setting in Hoogeboom et al. (2022) is suboptimal. For example, the conditional EDM on GEOM-Drug has a smaller number of parameters than the conditional EDM on QM9 (15M v.s. 26M), which is insufficient to capture the structure information. Nevertheless, our EEGSDE still improves the similarity by $\sim$%17. We provide generated molecules on GEOM-Drug in Appendix G.5.

Finally, we demonstrate that our EEGSDE is a flexible framework to generate molecules targeted to multiple properties, which is often the practical case. We additionally target to the quantum property $\alpha$ (polarizability) on QM9 by combining the energy function for structures in this section and the energy function for quantum properties in Section 5. Here we choose $\alpha = 100 \, \mathrm{Bohr}^3$, which is a relatively large value, and we expect it to encourage less isometrically shaped structures. As shown in Figure 3, the generated molecule aligns better with the target structure as the scaling factor $s_1$ grows, and meanwhile a ring substructure in the generated molecule vanishes as the scaling factor for polarizability $s_2$ grows, leading to a less isometrically shaped structure, which is as expected.

**Conclusion:** These results suggest that molecules generated by our EEGSDE align better with the target structure than molecules generated by the conditional EDM baseline. Besides, EEGSDE can generate molecules targeted to both specific structures and desired quantum properties by combining energy functions linearly. As a result, EEGSDE may benefit practical cases in molecular design when multiple properties should be considered at the same time.

## 7   CONCLUSION

This work presents equivariant energy-guided SDE (EEGSDE), a flexible framework for controllable 3D molecule generation under the guidance of an energy function in diffusion models. EEGSDE naturally exploits the geometric symmetry in 3D molecular conformation, as long as the energy function is invariant to orthogonal transformations. EEGSDE significantly improves the conditional EDM baseline in inverse molecular design targeted to quantum properties and molecular structures. Furthermore, EEGSDE is able to generate molecules with multiple target properties by combining the corresponding energy functions linearly.

## ACKNOWLEDGMENTS

We thank Han Guo and Zhen Jia for their help with their expertise in chemistry. This work was supported by NSF of China Projects (Nos. 62061136001, 61620106010, 62076145, U19B2034, U1811461, U19A2081, 6197222); Beijing Outstanding Young Scientist Program NO. BJJWZYJH012019100020098; a grant from Tsinghua Institute for Guo Qiang; the High Performance Computing Center, Tsinghua University; the Fundamental Research Funds for the Central Universities, and the Research Funds of Renmin University of China (22XNKJ13). J.Z was also supported by the XPlorer Prize. C. Li was also sponsored by Beijing Nova Program.

## ETHICS STATEMENT

Inverse molecular design is critical in fields like material science and drug discovery. Our EEGSDE is a flexible framework for inverse molecular design and thus might benefit these fields. Currently the negative consequences are not obvious.

## REPRODUCIBILITY STATEMENT

Our code is included in the supplementary material. The implementation of our experiment is described in Section 5 and Section 6. Further details such as the hyperparameters of training and model backbones are provided in Appendix F. We provide complete proofs and derivations of all theoretical results in Appendix A.

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

# A    DERIVATIONS

## A.1    ZERO COM SUBSPACE

The zero CoM subspace $X = \{\boldsymbol{x} \in \mathbb{R}^{Mn} : \sum_{i=1}^{M} \boldsymbol{x}^i = \boldsymbol{0}\}$ is a $(M-1)n$ dimensional subspace of $\mathbb{R}^{Mn}$. Therefore, there exists an isometric isomorphism $\phi$ from $\mathbb{R}^{(M-1)n}$ to $X$, i.e., $\phi$ is a linear bijection from $\mathbb{R}^{(M-1)n}$ to $X$, and $\|\phi(\hat{\boldsymbol{x}})\|_2 = \|\hat{\boldsymbol{x}}\|_2$ for all $\hat{\boldsymbol{x}} \in \mathbb{R}^{(M-1)n}$. We use $A_\phi \in \mathbb{R}^{Mn \times (M-1)n}$ represent the matrix corresponding to $\phi$, so we have $\phi(\hat{\boldsymbol{x}}) = A_\phi \hat{\boldsymbol{x}}$. An important property of the isometric isomorphism is that $A_\phi A_\phi^\top \boldsymbol{x} = \boldsymbol{x} - \overline{\boldsymbol{x}}$ for all $\boldsymbol{x} \in \mathbb{R}^{Mn}$. We show the proof as following.

**Proposition 1.** *Suppose $\phi$ is an isometric isomorphism from $\mathbb{R}^{(M-1)n}$ to $X$, and let $A_\phi \in \mathbb{R}^{Mn \times (M-1)n}$ be the matrix corresponding to $\phi$. Then we have $A_\phi A_\phi^\top \boldsymbol{x} = \boldsymbol{x} - \overline{\boldsymbol{x}}$ for all $\boldsymbol{x} \in \mathbb{R}^{Mn}$, where $\overline{\boldsymbol{x}} = \frac{1}{M} \sum_{i=1}^{M} \boldsymbol{x}^i$.*

*Proof.* We consider a new subspace of $\mathbb{R}^{Mn}$, $X^\perp = \{\boldsymbol{x} \in \mathbb{R}^{Mn} : \boldsymbol{x} \perp X\}$, i.e., the orthogonal component of $X$. We can verify that $X^\perp = \{\boldsymbol{x} \in \mathbb{R}^{Mn} : \boldsymbol{x}_1 = \boldsymbol{x}_2 = \cdots = \boldsymbol{x}_M\}$, and $X^\perp$ is $n$ dimensional. Thus, there exists an isometric isomorphism $\psi$ from $\mathbb{R}^n$ to $X^\perp$. Let $A_\psi \in \mathbb{R}^{Mn \times n}$ represent the matrix corresponding to $\psi$. Then we define $\lambda(\hat{\boldsymbol{x}}, \hat{\boldsymbol{y}}) = \phi(\hat{\boldsymbol{x}}) + \psi(\hat{\boldsymbol{y}})$, where $\hat{\boldsymbol{x}} \in \mathbb{R}^{(M-1)n}$ and $\hat{\boldsymbol{y}} \in \mathbb{R}^n$. The image of $\lambda$ is $\{\boldsymbol{x} + \boldsymbol{y} : \boldsymbol{x} \in X, \boldsymbol{y} \in X^\perp\} = \mathbb{R}^{Mn}$. Therefore, $\lambda$ is a linear bijection from $\mathbb{R}^{Mn}$ to $\mathbb{R}^{Mn}$, and the matrix corresponding to $\lambda$ is $A_\lambda = [A_\phi, A_\psi]$. Furthermore, $\lambda$ is an isometric isomorphism, since $\|\lambda(\hat{\boldsymbol{x}}, \hat{\boldsymbol{y}})\|^2 = \|\phi(\hat{\boldsymbol{x}})\|^2 + \|\psi(\hat{\boldsymbol{y}})\|^2 + 2\langle\phi(\hat{\boldsymbol{x}}), \psi(\hat{\boldsymbol{y}})\rangle = \|\hat{\boldsymbol{x}}\|^2 + \|\hat{\boldsymbol{y}}\|^2 = \|(\hat{\boldsymbol{x}}^\top, \hat{\boldsymbol{y}}^\top)^\top\|^2$. This means $A_\lambda$ is orthogonal transformation. Therefore, $A_\lambda A_\lambda^\top = A_\phi A_\phi^\top \boldsymbol{x} + A_\psi A_\psi^\top = \boldsymbol{I}$ and $A_\phi A_\phi^\top \boldsymbol{x} + A_\psi A_\psi^\top \boldsymbol{x} = \boldsymbol{x}$. Since $A_\phi A_\phi^\top \boldsymbol{x} \in X$ and $A_\psi A_\psi^\top \boldsymbol{x} \in X^\top$, we can conclude that $A_\phi A_\phi^\top \boldsymbol{x}$ is the orthogonal projection of $\boldsymbol{x}$ to $X$, which is exactly $\boldsymbol{x} - \overline{\boldsymbol{x}}$. □

Since $\mathbb{R}^{(M-1)n}$ and $X$ are two intrinsically equivalent spaces, an equivalence of distributions in these two spaces can also be established, as shown in the following propositions.

**Proposition 2.** *Suppose $\boldsymbol{x}$ is a random vector distributed in $X$, and $\hat{\boldsymbol{x}} = \phi^{-1}(\boldsymbol{x})$ is its equivalent representation in $\mathbb{R}^{(M-1)n}$. If $\boldsymbol{x} \sim q(\boldsymbol{x})$, then $\hat{\boldsymbol{x}} \sim \hat{q}(\hat{\boldsymbol{x}})$, where $\hat{q}(\hat{\boldsymbol{x}}) = q(\phi(\hat{\boldsymbol{x}}))$.*

**Proposition 3.** *Suppose $\boldsymbol{x}, \boldsymbol{y}$ is are two random vectors distributed in $X$, and $\hat{\boldsymbol{x}} = \phi^{-1}(\boldsymbol{x}), \hat{\boldsymbol{y}} = \phi^{-1}(\boldsymbol{y})$ are their equivalent representations in $\mathbb{R}^{(M-1)n}$. If $\boldsymbol{x}|\boldsymbol{y} \sim q(\boldsymbol{x}|\boldsymbol{y})$, then $\hat{\boldsymbol{x}}|\hat{\boldsymbol{y}} \sim \hat{q}(\hat{\boldsymbol{x}}|\hat{\boldsymbol{y}})$, where $\hat{q}(\hat{\boldsymbol{x}}|\hat{\boldsymbol{y}}) = q(\phi(\hat{\boldsymbol{x}})|\phi(\hat{\boldsymbol{y}})))$.*

## A.2    SDE IN THE PRODUCT SPACE

**Definition 1** (Gaussian distributions in the zero CoM subspace)**.** *Suppose $\boldsymbol{\mu} \in X$. Let $\mathcal{N}_X(\boldsymbol{x}|\boldsymbol{\mu}, \sigma^2) := (2\pi\sigma^2)^{-(M-1)n/2} \exp(-\frac{1}{2\sigma^2}\|\boldsymbol{x} - \boldsymbol{\mu}\|_2^2)$, which is the isotropic Gaussian distribution with mean $\boldsymbol{\mu}$ and variance $\sigma^2$ in the zero CoM subspace $X$.*

**Proposition 4** (Transition kernels of a SDE in the zero CoM subspace). *Suppose* $\mathrm{d}\boldsymbol{x} = f(t)\boldsymbol{x}\mathrm{d}t + g(t)\mathrm{d}\boldsymbol{w}_x$, $\boldsymbol{x}_0 \sim q(\boldsymbol{x}_0)$ *is a SDE in the zero CoM subspace. Then the transition kernel from* $\boldsymbol{x}_s$ *to* $\boldsymbol{x}_t$ $(0 \le s < t \le T)$ *can be expressed as* $q(\boldsymbol{x}_t|\boldsymbol{x}_s) = \mathcal{N}_X(\boldsymbol{x}_t|\sqrt{\alpha_{t|s}}\boldsymbol{x}_s, \beta_{t|s})$, *where* $\alpha_{t|s} = \exp(2\int_s^t f(\tau)\mathrm{d}\tau)$ *and* $\beta_{t|s} = \alpha_{t|s}\int_s^t g(\tau)^2/\alpha_{\tau|s}\mathrm{d}\tau$ *are two scalars determined by* $f(\cdot)$ *and* $g(\cdot)$.

*Proof.* Firstly, we map the process $\{\boldsymbol{x}_t\}_{t=0}^T$ in the zero CoM subspace to the equivalent space $\mathbb{R}^{(M-1)n}$ through the isometric isomorphism $\phi$ introduced in Appendix A.1. This produces a new process $\{\hat{\boldsymbol{x}}_t\}_{t=0}^T$, where $\hat{\boldsymbol{x}}_t = \phi^{-1}(\boldsymbol{x}_t)$. By applying $\phi^{-1}$ to the SDE in the zero CoM subspace, we know $\hat{\boldsymbol{x}}_t = \phi^{-1}(\boldsymbol{x}_t)$ satisfies the following SDE in $\mathbb{R}^{(M-1)n}$:

$$\mathrm{d}\hat{\boldsymbol{x}} = f(t)\hat{\boldsymbol{x}}\mathrm{d}t + g(t)\mathrm{d}\hat{\boldsymbol{w}}, \quad \hat{\boldsymbol{x}}_0 \sim \hat{q}(\hat{\boldsymbol{x}}_0), \tag{7}$$

where $\hat{\boldsymbol{w}}$ is the standard Wiener process in $\mathbb{R}^{(M-1)n}$ and $\hat{q}(\hat{\boldsymbol{x}}_0) = q(\phi(\hat{\boldsymbol{x}}_0))$. According to Song et al. (2020), the transition of Eq. (7) from $\hat{\boldsymbol{x}}_s$ to $\hat{\boldsymbol{x}}_t$ $(s < t)$ can be expressed as $\hat{q}(\hat{\boldsymbol{x}}_t|\hat{\boldsymbol{x}}_s) = \mathcal{N}(\hat{\boldsymbol{x}}_t|\sqrt{\alpha_{t|s}}\hat{\boldsymbol{x}}_s, \beta_{t|s}\boldsymbol{I})$, where $\alpha_{t|s} = \exp(2\int_s^t f(\tau)\mathrm{d}\tau)$ and $\beta_{t|s} = \alpha_{t|s}\int_s^t g(\tau)^2/\alpha_{\tau|s}\mathrm{d}\tau$ are two scalars determined by $f(\cdot)$ and $g(\cdot)$. According to Proposition 3, we know the transition kernel $q(\boldsymbol{x}_t|\boldsymbol{x}_s)$ from $\boldsymbol{x}_s$ to $\boldsymbol{x}_t$ satisfies

$$q(\boldsymbol{x}_t|\boldsymbol{x}_s) = \hat{q}(\phi^{-1}(\boldsymbol{x}_t)|\phi^{-1}(\boldsymbol{x}_s)) = (2\pi\beta_{t|s})^{-(M-1)n/2}\exp(-\frac{1}{2\beta_{t|s}}\|\phi^{-1}(\boldsymbol{x}_t) - \sqrt{\alpha_{t|s}}\phi^{-1}(\boldsymbol{x}_s)\|_2^2)$$

$$= (2\pi\beta_{t|s})^{-(M-1)n/2}\exp(-\frac{1}{2\beta_{t|s}}\|\phi^{-1}(\boldsymbol{x}_t - \sqrt{\alpha_{t|s}}\boldsymbol{x}_s)\|_2^2) \qquad \text{// linearity of } \phi^{-1}$$

$$= (2\pi\beta_{t|s})^{-(M-1)n/2}\exp(-\frac{1}{2\beta_{t|s}}\|\boldsymbol{x}_t - \sqrt{\alpha_{t|s}}\boldsymbol{x}_s\|_2^2) \qquad \text{// norm preserving of } \phi^{-1}$$

$$= \mathcal{N}_X(\boldsymbol{x}_t|\sqrt{\alpha_{t|s}}\boldsymbol{x}_s, \beta_{t|s}).$$

$\square$

**Proposition 5** (Transition kernels of the SDE in the product space). *Suppose* $\mathrm{d}\boldsymbol{z} = f(t)\boldsymbol{z}\mathrm{d}t + g(t)\mathrm{d}(\boldsymbol{w}_x, \boldsymbol{w}_h)$, $\boldsymbol{z}_0 \sim q(\boldsymbol{z}_0)$ *is the SDE in the product space* $X \times \mathbb{R}^{Md}$, *as introduced in Eq. (3). Then the transition kernel from* $\boldsymbol{z}_s$ *to* $\boldsymbol{z}_t$ $(0 \le s < t \le T)$ *can be expressed as* $q(\boldsymbol{z}_t|\boldsymbol{z}_s) = q(\boldsymbol{x}_t|\boldsymbol{x}_s)q(\boldsymbol{h}_t|\boldsymbol{h}_s)$, *where* $q(\boldsymbol{x}_t|\boldsymbol{x}_s) = \mathcal{N}_X(\boldsymbol{x}_t|\sqrt{\alpha_{t|s}}\boldsymbol{x}_s, \beta_{t|s})$ *and* $q(\boldsymbol{h}_t|\boldsymbol{h}_s) = \mathcal{N}(\boldsymbol{h}_t|\sqrt{\alpha_{t|s}}\boldsymbol{h}_s, \beta_{t|s})$. *Here* $\alpha_{t|s}$ *and* $\beta_{t|s}$ *are defined as in Proposition 4.*

*Proof.* Since $\boldsymbol{w}_x$ and $\boldsymbol{w}_h$ are independent to each other, the transition kernel from $\boldsymbol{z}_s$ to $\boldsymbol{z}_t$ can be factorized as $q(\boldsymbol{z}_t|\boldsymbol{z}_s) = q(\boldsymbol{x}_t|\boldsymbol{x}_s)q(\boldsymbol{h}_t|\boldsymbol{h}_s)$, where $q(\boldsymbol{x}_t|\boldsymbol{x}_s)$ is the transition kernel of $\mathrm{d}\boldsymbol{x} = f(t)\boldsymbol{x}\mathrm{d}t + g(t)\mathrm{d}\boldsymbol{w}_x$ and $q(\boldsymbol{h}_t|\boldsymbol{h}_s)$ is the transition kernel of $\mathrm{d}\boldsymbol{h} = f(t)\boldsymbol{h}\mathrm{d}t + g(t)\mathrm{d}\boldsymbol{w}_h$. According to Proposition 4 and Song et al. (2020), we have $q(\boldsymbol{x}_t|\boldsymbol{x}_s) = \mathcal{N}_X(\boldsymbol{x}_t|\sqrt{\alpha_{t|s}}\boldsymbol{x}_s, \beta_{t|s})$ and $q(\boldsymbol{h}_t|\boldsymbol{h}_s) = \mathcal{N}(\boldsymbol{h}_t|\sqrt{\alpha_{t|s}}\boldsymbol{h}_s, \beta_{t|s})$. $\square$

**Remark 1.** *The marginal distribution of* $\boldsymbol{z}_t$ *is*

$$q_t(\boldsymbol{z}_t) = \int_Z \int_{\mathbb{R}^{Md}} q(\boldsymbol{z}_0)q(\boldsymbol{x}_t|\boldsymbol{x}_0)q(\boldsymbol{h}_t|\boldsymbol{h}_0)\mathrm{d}\boldsymbol{h}_0\lambda(\mathrm{d}\boldsymbol{x}_0),$$

*where* $\lambda$ *is the Lebesgue measure in the zero CoM subspace* $X$. *While* $q_t(\boldsymbol{z}_t)$ *is a distribution in* $X \times \mathbb{R}^{Md}$, *it has a natural differentiable extension to* $\mathbb{R}^{Mn} \times \mathbb{R}^{Md}$, *since* $q(\boldsymbol{x}_t|\boldsymbol{x}_0) = \mathcal{N}_X(\boldsymbol{x}_t|\sqrt{\alpha_{t|s}}\boldsymbol{x}_s, \beta_{t|s})$ *has a differentiable extension to* $\mathbb{R}^{Mn}$ *according to Definition 1. Thus, we can take gradient of* $q_t(\boldsymbol{z}_t)$ *w.r.t.* $\boldsymbol{x}_t$ *in the whole space* $\mathbb{R}^{Mn}$.

**Proposition 6** (Time reversal of the SDE in the product space). *Suppose* $\mathrm{d}\boldsymbol{z} = f(t)\boldsymbol{z}\mathrm{d}t + g(t)\mathrm{d}(\boldsymbol{w}_x, \boldsymbol{w}_h)$, $\boldsymbol{z}_0 \sim q(\boldsymbol{z}_0)$ *is the SDE in the product space* $X \times \mathbb{R}^{Md}$, *as introduced in Eq. (3). Then its time reversal satisfies the following reverse-time SDE, which can be represented by both the score function form and the noise prediction form:*

$$\mathrm{d}\boldsymbol{z} = [f(t)\boldsymbol{z} - g(t)^2 \underbrace{(\nabla_{\boldsymbol{x}}\log q_t(\boldsymbol{z}) - \overline{\nabla_{\boldsymbol{x}}\log q_t(\boldsymbol{z})}, \nabla_{\boldsymbol{h}}\log q_t(\boldsymbol{z}))}_{\text{score function form}}]\mathrm{d}t + g(t)\mathrm{d}(\tilde{\boldsymbol{w}}_x, \tilde{\boldsymbol{w}}_h),$$

$$= [f(t)\boldsymbol{z} + \frac{g(t)^2}{\sqrt{\beta_{t|0}}} \underbrace{\mathbb{E}_{q(\boldsymbol{z}_0|\boldsymbol{z}_t)}\boldsymbol{\epsilon}_t}_{\text{noise prediction form}}]\mathrm{d}t + g(t)\mathrm{d}(\tilde{\boldsymbol{w}}_x, \tilde{\boldsymbol{w}}_h), \quad \boldsymbol{z}_T \sim q_T(\boldsymbol{z}_T),$$

where $\tilde{w}_x$ and $\tilde{w}_h$ are reverse-time standard Wiener processes in $X$ and $\mathbb{R}^{Md}$ respectively, and $\epsilon_t = \frac{z_t - \sqrt{\alpha_{t|0}}z_0}{\sqrt{\beta_{t|0}}}$ is the standard Gaussian noise in $X \times \mathbb{R}^{Md}$ injected to $z_0$.

*Furthermore, we have* $\nabla_x \log q_t(x) - \overline{\nabla_x \log q_t(x)} = -\frac{1}{\sqrt{\beta_{t|0}}}\mathbb{E}_{q(x_0|x_t)}\epsilon_t$, *where* $\epsilon_t = \frac{x_t - \sqrt{\alpha_{t|0}}x_0}{\sqrt{\beta_{t|0}}}$ *is the standard Gaussian noise in the zero CoM subspace injected to* $x_0$.

*Proof.* Let $\hat{z}_t = (\hat{x}_t, h_t)$, where $\hat{x}_t = \phi^{-1}(x_t)$, introduced in the proof of Proposition 4. Then $\{\hat{z}_t\}_{t=0}^T$ is a process in $\mathbb{R}^{(M-1)n} \times \mathbb{R}^{Md}$ determined by the following SDE

$$d\hat{z} = f(t)\hat{z}dt + g(t)d\hat{w}, \quad \hat{z}_0 \sim \hat{q}(\hat{z}_0), \tag{8}$$

where $\hat{w}$ is the standard Wiener process in $\mathbb{R}^{(M-1)n} \times \mathbb{R}^{Md}$ and $\hat{q}(\hat{z}_0) = q(\phi(\hat{x}_0), h)$.

According to Song et al. (2020), Eq. (8) has a time reversal:

$$d\hat{z} = [f(t)\hat{z} - g(t)^2 \nabla_{\hat{z}} \log \hat{q}_t(\hat{z})]dt + g(t)d\tilde{w}, \quad \hat{z}_T \sim \hat{q}_T(\hat{z}_T), \tag{9}$$

where $\hat{q}_t(z)$ is the marginal distribution of $\hat{z}_t$, which satisfies $\hat{q}_t(\hat{z}_t) = q_t(\phi(\hat{x}_t), h_t)$ according to Proposition 2, and $\tilde{w}$ is the reverse-time standard Wiener process in $\mathbb{R}^{(M-1)n} \times \mathbb{R}^{Md}$.

Then we apply the linear transformation $T\hat{z} = (\phi(\hat{x}), h)$ to Eq. (9), which maps $\hat{z}_t$ back to $z_t$. This yields

$$dz = [f(t)z - g(t)^2 T(\nabla_{\hat{z}} \log \hat{q}_t(\hat{z}))]dt + g(t)d(\tilde{w}_x, \tilde{w}_h), \quad z_T \sim q_T(z_T). \tag{10}$$

Here $\tilde{w}_x$ and $\tilde{w}_h$ are reverse-time standard Wiener processes in $X$ and $\mathbb{R}^{Md}$ respectively, and $T(\nabla_{\hat{z}} \log \hat{q}_t(\hat{z}))$ can be expressed as $(\phi(\nabla_{\hat{x}} \log \hat{q}_t(\hat{z})), \nabla_h \log \hat{q}_t(\hat{z}))$.

Since $\nabla_{\hat{x}} \log \hat{q}_t(\hat{z}) = A_\phi^\top \nabla_x \log q_t(x, h) = A_\phi^\top \nabla_x \log q_t(z)$, we have $\phi(\nabla_{\hat{x}} \log \hat{q}_t(\hat{z})) = A_\phi A_\phi^\top \nabla_x \log q_t(z)$, where $A_\phi$ represents the matrix corresponding to $\phi$. According to Proposition 1, we have $\phi(\nabla_{\hat{x}} \log \hat{q}_t(\hat{z})) = \nabla_x \log q_t(z) - \overline{\nabla_x \log q_t(z)}$. Besides, $\nabla_h \log \hat{q}_t(\hat{z}) = \nabla_h \log q_t(z)$. Thus, Eq. (10) can be written as

$$dz = [f(t)z - g(t)^2(\nabla_x \log q_t(z) - \overline{\nabla_x \log q_t(z)}, \nabla_h \log q_t(z))]dt + g(t)d(\tilde{w}_x, \tilde{w}_h),$$

which is the score function form of the reverse-time SDE.

We can also write the score function in Eq. (9) as $\nabla_{\hat{z}} \log \hat{q}_t(\hat{z}) = \mathbb{E}_{\hat{q}(\hat{z}_0|\hat{z}_t)} \nabla_{\hat{z}} \log \hat{q}(\hat{z}_t|\hat{z}_0) = -\frac{1}{\sqrt{\beta_{t|0}}}\mathbb{E}_{\hat{q}(\hat{z}_0|\hat{z}_t)}\hat{\epsilon}_t$, where $\hat{\epsilon}_t = \frac{\hat{z}_t - \sqrt{\alpha_{t|0}}\hat{z}_0}{\sqrt{\beta_{t|0}}}$ is the standard Gaussian noise injected to $\hat{z}_0$. With this expression, we have $T(\nabla_{\hat{z}} \log \hat{q}_t(\hat{z})) = -\frac{1}{\sqrt{\beta_{t|0}}}\mathbb{E}_{\hat{q}(\hat{z}_0|\hat{z}_t)}T(\hat{\epsilon}_t) = -\frac{1}{\sqrt{\beta_{t|0}}}\mathbb{E}_{q(z_0|z_t)}\epsilon_t$, where $\epsilon_t = \frac{z_t - \sqrt{\alpha_{t|0}}z_0}{\sqrt{\beta_{t|0}}}$ is the standard Gaussian noise in $X \times \mathbb{R}^{Md}$ injected to $z_0$. Thus, Eq. (10) can also be written as

$$dz = [f(t)z + \frac{g(t)^2}{\sqrt{\beta_{t|0}}}\mathbb{E}_{q(z_0|z_t)}\epsilon_t]dt + g(t)d(\tilde{w}_x, \tilde{w}_h),$$

which is the noise prediction form of the reverse-time SDE.

$\square$

## A.3 EQUIVARIANCE

**Theorem 1.** *Let* $(\epsilon_\theta^x(z_t, t), \epsilon_\theta^h(z_t, t)) = \epsilon_\theta(z_t, t)$, *where* $\epsilon_\theta^x(z_t, t)$ *and* $\epsilon_\theta^h(z_t, t)$ *are the predicted noise of* $x_t$ *and* $h_t$ *respectively. If for any orthogonal transformation* $R \in \mathbb{R}^{n \times n}$, $\epsilon_\theta(z_t, t)$ *is equivariant to* $R$, *i.e.,* $\epsilon_\theta(Rx_t, h_t, t) = (R\epsilon_\theta^x(x_t, h_t, t), \epsilon_\theta^h(x_t, h_t, t))$, *and* $p_T(z_T)$ *is invariant to* $R$, *i.e.,* $p_T(Rx_T, h_T) = p_T(x_T, h_T)$, *then* $p_\theta(z_0)$ *is invariant to any rotational transformation.*

*Proof.* Suppose $R \in \mathbb{R}^{n \times n}$ is an orthogonal transformation. Let $z_t^\theta = (x_t^\theta, h_t^\theta)$ $(0 \le t \le T)$ be the process determined by Eq. (5). Let $y_t^\theta = Rx_t^\theta$ and $u_t^\theta = (y_t^\theta, h_t^\theta)$. We use $p_t^u(u_t)$ and $p_t^z(z_t)$ to denote the distributions of $u_t^\theta$ and $z_t^\theta$ respectively, and they satisfy $p_t^u(y_t, h_t) = p_t^z(R^{-1}y_t, h_t)$.

By applying the transformation $T\boldsymbol{z} = (\boldsymbol{R}\boldsymbol{x}, \boldsymbol{h})$ to Eq. (5), we know the new process $\{\boldsymbol{u}_t^{\boldsymbol{\theta}}\}_{t=0}^T$ satisfies the following SDE:

$$\mathrm{d}\boldsymbol{u} = T\mathrm{d}\boldsymbol{z} = [f(t)\boldsymbol{u} + \frac{g(t)^2}{\sqrt{\beta_{t|0}}}(\boldsymbol{R}\boldsymbol{\epsilon}_{\boldsymbol{\theta}}^x(\boldsymbol{z}, t), \boldsymbol{\epsilon}_{\boldsymbol{\theta}}^h(\boldsymbol{z}, t))]\mathrm{d}t + g(t)\mathrm{d}(\boldsymbol{R}\tilde{\boldsymbol{w}}_x, \tilde{\boldsymbol{w}}_h)$$

$$= [f(t)\boldsymbol{u} + \frac{g(t)^2}{\sqrt{\beta_{t|0}}}(\boldsymbol{\epsilon}_{\boldsymbol{\theta}}^x(\boldsymbol{R}\boldsymbol{x}, \boldsymbol{h}, t), \boldsymbol{\epsilon}_{\boldsymbol{\theta}}^h(\boldsymbol{R}\boldsymbol{x}, \boldsymbol{h}, t))]\mathrm{d}t + g(t)\mathrm{d}(\boldsymbol{R}\tilde{\boldsymbol{w}}_x, \tilde{\boldsymbol{w}}_h) \quad \text{// by equivariance}$$

$$= [f(t)\boldsymbol{u} + \frac{g(t)^2}{\sqrt{\beta_{t|0}}}\boldsymbol{\epsilon}_{\boldsymbol{\theta}}(\boldsymbol{u}, t)]\mathrm{d}t + g(t)\mathrm{d}(\tilde{\boldsymbol{w}}_x, \tilde{\boldsymbol{w}}_h), \quad \text{// Winner process is isotropic}$$

and its initial distribution is $p_T^{\boldsymbol{u}}(\boldsymbol{u}_T) = p_T^{\boldsymbol{u}}(\boldsymbol{y}_T, \boldsymbol{h}_T) = p_T^{\boldsymbol{z}}(\boldsymbol{R}^{-1}\boldsymbol{y}_T, \boldsymbol{h}_T) = p_T(\boldsymbol{R}^{-1}\boldsymbol{y}_T, \boldsymbol{h}_T) = p_T(\boldsymbol{y}_T, \boldsymbol{h}_T) = p_T(\boldsymbol{u}_T)$. Thus, the SDE of $\{\boldsymbol{u}_t^{\boldsymbol{\theta}}\}_{t=0}^T$ is exactly the same to that of $\{\boldsymbol{z}_t^{\boldsymbol{\theta}}\}_{t=0}^T$. This indicates that the distribution of $\boldsymbol{u}_0^{\boldsymbol{\theta}}$ is the same to the distribution of $\boldsymbol{z}_0^{\boldsymbol{\theta}}$, i.e., $p_0^{\boldsymbol{u}}(\boldsymbol{u}_0) = p_0^{\boldsymbol{z}}(\boldsymbol{u}_0) = p_{\boldsymbol{\theta}}(\boldsymbol{u}_0)$. Also note that $p_0^{\boldsymbol{u}}(\boldsymbol{u}_0) = p_0^{\boldsymbol{z}}(\boldsymbol{R}^{-1}\boldsymbol{y}_0, \boldsymbol{h}_0) = p_{\boldsymbol{\theta}}(\boldsymbol{R}^{-1}\boldsymbol{y}_0, \boldsymbol{h}_0)$. Thus, $p_{\boldsymbol{\theta}}(\boldsymbol{u}_0) = p_{\boldsymbol{\theta}}(\boldsymbol{R}^{-1}\boldsymbol{y}_0, \boldsymbol{h}_0)$, and consequently $p_{\boldsymbol{\theta}}(\boldsymbol{R}\boldsymbol{x}_0, \boldsymbol{h}_0) = p_{\boldsymbol{\theta}}(\boldsymbol{x}_0, \boldsymbol{h}_0)$. This means $p_{\boldsymbol{\theta}}(\boldsymbol{z}_0)$ is invariant to any orthogonal transformation, which includes rotational transformations as special cases. □

**Theorem 2.** *Suppose the assumptions in Theorem 1 hold and $E(\boldsymbol{z}, c, t)$ is invariant to any orthogonal transformation $\boldsymbol{R}$, i.e., $E(\boldsymbol{R}\boldsymbol{x}, \boldsymbol{h}, c, t) = E(\boldsymbol{x}, \boldsymbol{h}, c, t)$. Then $p_{\boldsymbol{\theta}}(\boldsymbol{z}_0|c)$ is invariant to any rotational transformation.*

*Proof.* Suppose $\boldsymbol{R} \in \mathbb{R}^{n \times n}$ is an orthogonal transformation. Taking gradient to both side of $E(\boldsymbol{R}\boldsymbol{x}, \boldsymbol{h}, c, t) = E(\boldsymbol{x}, \boldsymbol{h}, c, t)$ w.r.t. $\boldsymbol{x}$, we get

$$\boldsymbol{R}^\top \nabla_{\boldsymbol{y}} E(\boldsymbol{y}, \boldsymbol{h}, c, t)|_{\boldsymbol{y}=\boldsymbol{R}\boldsymbol{x}} = \nabla_{\boldsymbol{x}} E(\boldsymbol{x}, \boldsymbol{h}, c, t).$$

Multiplying $\boldsymbol{R}$ to both sides, we get

$$\nabla_{\boldsymbol{y}} E(\boldsymbol{y}, \boldsymbol{h}, c, t)|_{\boldsymbol{y}=\boldsymbol{R}\boldsymbol{x}} = \boldsymbol{R}\nabla_{\boldsymbol{x}} E(\boldsymbol{x}, \boldsymbol{h}, c, t).$$

Let $\boldsymbol{\phi}(\boldsymbol{z}, c, t) = (\nabla_{\boldsymbol{x}} E(\boldsymbol{z}, c, t) - \overline{\nabla_{\boldsymbol{x}} E(\boldsymbol{z}, c, t)}, \nabla_{\boldsymbol{h}} E(\boldsymbol{z}, c, t))$. Then we have

$$\boldsymbol{\phi}(\boldsymbol{R}\boldsymbol{x}, \boldsymbol{h}, c, t) = (\nabla_{\boldsymbol{y}} E(\boldsymbol{y}, \boldsymbol{h}, c, t) - \overline{\nabla_{\boldsymbol{y}} E(\boldsymbol{y}, \boldsymbol{h}, c, t)}, \nabla_{\boldsymbol{h}} E(\boldsymbol{y}, \boldsymbol{h}, c, t))|_{\boldsymbol{y}=\boldsymbol{R}\boldsymbol{x}}$$

$$= (\boldsymbol{R}\nabla_{\boldsymbol{x}} E(\boldsymbol{x}, \boldsymbol{h}, c, t) - \boldsymbol{R}\overline{\nabla_{\boldsymbol{x}} E(\boldsymbol{x}, \boldsymbol{h}, c, t)}, \nabla_{\boldsymbol{h}} E(\boldsymbol{R}\boldsymbol{x}, \boldsymbol{h}, c, t))$$

$$= (\boldsymbol{R}(\nabla_{\boldsymbol{x}} E(\boldsymbol{x}, \boldsymbol{h}, c, t) - \overline{\nabla_{\boldsymbol{x}} E(\boldsymbol{x}, \boldsymbol{h}, c, t)}), \nabla_{\boldsymbol{h}} E(\boldsymbol{x}, \boldsymbol{h}, c, t)).$$

Thus, $\boldsymbol{\phi}(\boldsymbol{z}, c, t)$ is equivariant to $\boldsymbol{R}$. Let $\hat{\boldsymbol{\epsilon}}_{\boldsymbol{\theta}}(\boldsymbol{z}, c, t) = \boldsymbol{\epsilon}_{\boldsymbol{\theta}}(\boldsymbol{z}, t) + \sqrt{\beta_{t|0}}\boldsymbol{\phi}(\boldsymbol{z}, c, t)$, which is a linear combination of two equivariant functions and is also equivariant to $\boldsymbol{R}$.

Then, Eq. (6) can be written as

$$\mathrm{d}\boldsymbol{z} = [f(t)\boldsymbol{z} + \frac{g(t)^2}{\sqrt{\beta_{t|0}}}\hat{\boldsymbol{\epsilon}}_{\boldsymbol{\theta}}(\boldsymbol{z}, c, t)]\mathrm{d}t + g(t)\mathrm{d}(\tilde{\boldsymbol{w}}_x, \tilde{\boldsymbol{w}}_h), \quad \boldsymbol{z}_T \sim p_T(\boldsymbol{z}_T).$$

According to Theorem 1, we know its marginal distribution at time $t = 0$, i.e., $p_{\boldsymbol{\theta}}(\boldsymbol{z}_0|c)$, is invariant to any rotational transformation.

□

## B  SAMPLING

In Algorithm 1, we present the Euler-Maruyama method to sample from EEGSDE in Eq. 6.

---

**Algorithm 1** Sample from EEGSDE using the Euler-Maruyama method

---

**Require:** Number of steps $N$
  $\Delta t = \frac{T}{N}$
  $\boldsymbol{z} \leftarrow (\boldsymbol{x} - \overline{\boldsymbol{x}}, \boldsymbol{h})$, where $\boldsymbol{x} \sim \mathcal{N}(\boldsymbol{0}, 1), \boldsymbol{h} \sim \mathcal{N}(\boldsymbol{0}, 1)$          {Sample from the prior $p_T(\boldsymbol{z}_T)$}
  **for** $i = N$ to 1 **do**
    $t \leftarrow i\Delta t$
    $\boldsymbol{g}_x \leftarrow \nabla_{\boldsymbol{x}} E(\boldsymbol{z}, c, t), \boldsymbol{g}_h \leftarrow \nabla_{\boldsymbol{h}} E(\boldsymbol{z}, c, t)$       {Calculate the gradient of the energy function}
    $\boldsymbol{g} \leftarrow (\boldsymbol{g}_x - \overline{\boldsymbol{g}_x}, \boldsymbol{g}_h)$                                       {Subtract the CoM of the gradient}
    $\boldsymbol{F} \leftarrow f(t)\boldsymbol{z} + g(t)^2(\frac{1}{\sqrt{\beta_{t|0}}}\boldsymbol{\epsilon_\theta}(\boldsymbol{z}, t) + \boldsymbol{g})$
    $\boldsymbol{\epsilon} \leftarrow (\boldsymbol{\epsilon}_x - \overline{\boldsymbol{\epsilon}_x}, \boldsymbol{\epsilon}_h)$, where $\boldsymbol{\epsilon}_x \sim \mathcal{N}(\boldsymbol{0}, 1), \boldsymbol{\epsilon}_h \sim \mathcal{N}(\boldsymbol{0}, 1)$
    $\boldsymbol{z} \leftarrow \boldsymbol{z} - \boldsymbol{F}\Delta t + g(t)\sqrt{\Delta t}\boldsymbol{\epsilon}$                                  {Update $\boldsymbol{z}$ according to Eq. (6)}
  **end for**
  **return** $\boldsymbol{z}$

---

## C   Conditional Noise Prediction Networks

In Eq. (6), we can alternatively use a conditional noise prediction network $\boldsymbol{\epsilon_\theta}(\boldsymbol{z}, c, t)$ for a stronger guidance, as follows

$$
\begin{aligned}
\mathrm{d}\boldsymbol{z} = [f(t)\boldsymbol{z} + g(t)^2(\frac{1}{\sqrt{\beta_{t|0}}}\boldsymbol{\epsilon_\theta}(\boldsymbol{z}, c, t) \\
+ (\nabla_{\boldsymbol{x}} E(\boldsymbol{z}, c, t) - \overline{\nabla_{\boldsymbol{x}} E(\boldsymbol{z}, c, t)}, \nabla_{\boldsymbol{h}} E(\boldsymbol{z}, c, t)))]\mathrm{d}t + g(t)\mathrm{d}(\tilde{\boldsymbol{w}}_x, \tilde{\boldsymbol{w}}_h), \ \boldsymbol{z}_T \sim p_T(\boldsymbol{z}_T).
\end{aligned}
$$

The conditional noise prediction network is trained similarly to the unconditional one, using the following MSE loss

$$
\min_{\boldsymbol{\theta}} \mathbb{E}_t \mathbb{E}_{q(c, \boldsymbol{z}_0, \boldsymbol{z}_t)} w(t) \|\boldsymbol{\epsilon_\theta}(\boldsymbol{z}_t, c, t) - \boldsymbol{\epsilon}_t\|^2.
$$

## D   Parameterization of Noise Prediction Networks

We parameterize the noise prediction network following Hoogeboom et al. (2022), and we provide the specific parameterization for completeness. For the unconditional model $\boldsymbol{\epsilon_\theta}(\boldsymbol{z}, t)$, we first concatenate each atom feature $\boldsymbol{h}^i$ and $t$, which gives $\boldsymbol{h}^{i\prime} = (\boldsymbol{h}^i, t)$. Then we input $\boldsymbol{x}$ and $\boldsymbol{h}' = (\boldsymbol{h}^{1\prime}, \dots, \boldsymbol{h}^{M\prime})$ to the EGNN as follows

$$
(\boldsymbol{a}^x, \boldsymbol{a}^{h\prime}) = \mathrm{EGNN}(\boldsymbol{x}, \boldsymbol{h}') - (\boldsymbol{x}, \boldsymbol{0}).
$$

Finally, we subtract the CoM of $\boldsymbol{a}^x$, and gets the parameterization of $\boldsymbol{\epsilon_\theta}(\boldsymbol{z}, t)$:

$$
\boldsymbol{\epsilon_\theta}(\boldsymbol{z}, t) = (\boldsymbol{a}^x - \overline{\boldsymbol{a}^x}, \boldsymbol{a}^h),
$$

where $\boldsymbol{a}^h$ comes from discarding the last component of $\boldsymbol{a}^{h\prime}$ that corresponds to the time.

For the conditional model $\boldsymbol{\epsilon_\theta}(\boldsymbol{z}, c, t)$, we additionally concatenate $c$ to the atom feature $\boldsymbol{h}^i$, i.e., $\boldsymbol{h}^{i\prime} = (\boldsymbol{h}^i, t, c)$, and other parts in the parameterization remain the same.

## E   Details of Energy Functions

### E.1   Parameterization of Time-Dependent Models

The time-dependent property prediction model $g(\boldsymbol{z}_t, t)$ is parameterized using the second component in the output of EGNN (see Section 3) followed by a decoder (Dec):

$$
g(\boldsymbol{z}_t, t) = \mathrm{Dec}(\mathrm{EGNN}^h(\boldsymbol{x}_t, \boldsymbol{h}'_t)), \quad \boldsymbol{h}'_t = \mathrm{concatenate}(\boldsymbol{h}_t, t), \tag{11}
$$

where the concatenation is performed on each atom feature, and the decoder is a small neural network based on Satorras et al. (2021b). This parameterization ensures that the energy function

$E(\boldsymbol{z}_t, c, t)$ is invariant to orthogonal transformations, and thus the distribution of generated samples is also invariant according to Theorem 2.

Similarly, the time-dependent multi-label classifier is parameterized by EGNN as

$$m(\boldsymbol{z}_t, t) = \sigma(\text{Dec}(\text{EGNN}^h(\boldsymbol{x}_t, \boldsymbol{h}'_t))), \quad \boldsymbol{h}'_t = \text{concatenate}(\boldsymbol{h}_t, t).$$

The multi-label classifier has the same backbone to the property prediction model in Eq. (11), except that the decoder outputs a vector of dimension $L$, and the sigmoid function $\sigma$ is adopted for multi-label classification. Similarly to Eq. (11), the EGNN in the multi-label classifier guarantees the invariance of the distribution of generated samples according to Theorem 2.

### E.2 TRAINING OBJECTIVES OF ENERGY FUNCTIONS

**Time-dependent property prediction model.** Since the quantum property is a scalar, we train the time-dependent property prediction model $g(\boldsymbol{z}_t, t)$ using the $\ell_1$ loss

$$\mathbb{E}_t \mathbb{E}_{q(c, \boldsymbol{z}_0, \boldsymbol{z}_t)} |g(\boldsymbol{z}_t, t) - c|,$$

where $t$ is uniformly sampled from $[0, T]$.

**Time-dependent multi-label classifier.** Since the fingerprint is a bit map, predicting it can be viewed as a multi-label classification task. Thus, we use a time dependent multi-label classifier $m(\boldsymbol{z}_t, t)$, and train it using the binary cross entropy loss

$$\mathbb{E}_t \mathbb{E}_{q(c, \boldsymbol{x}_0, \boldsymbol{x}_t)} \sum_{l=1}^{L} c_l \log m_l(\boldsymbol{z}_t, t) + (1 - c_l) \log(1 - m_l(\boldsymbol{z}_t, t)),$$

where $t$ is uniformly sampled from $[0, T]$, and $m_l(\boldsymbol{z}_t, t)$ is the $l$-th component of $m(\boldsymbol{z}_t, t)$.

## F EXPERIMENTAL DETAILS

### F.1 HOW TO GENERATE MOLECULES TARGETED TO MULTIPLE QUANTUM PROPERTIES

When we want to generate molecules with $K$ quantum properties $\boldsymbol{c} = (c_1, c_2, \ldots, c_K)$, we combine energy functions for single properties linearly as $E(\boldsymbol{z}_t, \boldsymbol{c}, t) = \sum_{k=1}^{K} E_k(\boldsymbol{z}_t, c_k, t)$, where $E_k(\boldsymbol{z}_t, c_k, t) = s_k |g_k(\boldsymbol{z}_t, t) - c_k|^2$ is the energy function for the $k$-th property, $s_k$ is the scaling factor and $g_k(\boldsymbol{z}_t, t)$ is the time-dependent property prediction model for the $k$-th property. Then we use the gradient of $E(\boldsymbol{z}_t, \boldsymbol{c}, t)$ to guide the reverse SDE as described in Eq. (6).

### F.2 THE "U-BOUND" AND "#ATOMS" BASELINES

The "U-bound" and "#Atoms" baselines are from Hoogeboom et al. (2022). The "U-bound" baseline shuffles the labels in $D_b$ and then calculate the loss of $\phi_c$ on it, which can be regarded as an upper bound of the MAE. The "#Atoms" baseline predicts a quantum property $c$ using only the number of atoms in a molecule.

### F.3 GENERATING MOLECULES WITH DESIRED QUANTUM PROPERTIES

For the noise prediction network, we use the same setting with EDM (Hoogeboom et al., 2022) for a fair comparison, where the models is trained $\sim$2000 epochs with a batch size of 64, a learning rate of 0.0001 with the Adam optimizer and an exponential moving average (EMA) with a rate of 0.9999.

The EGNN used in the energy function has 192 hidden features and 7 layers. We train 2000 epochs with a batch size of 128, a learning rate of 0.0001 with the Adam optimizer and an exponential moving average (EMA) with a rate of 0.9999.

During evaluation, we need to generate a set of molecules. Following the EDM (Hoogeboom et al., 2022), we firstly sample the number of atoms in a molecule $M \sim p(M)$ and the property value $c \sim p(c|M)$ (or $c_1 \sim p(c_1|M), c_2 \sim p(c_2|M), \ldots, c_K \sim p(c_K|M)$ for multiple properties). Here $p(M)$ is the distribution of molecule sizes on training data, and $p(c|M)$ is the distribution of the property on training data. Then we generate a molecule given $M, c$.

### F.4 GENERATING MOLECULES WITH TARGET STRUCTURES

**Computation of Tanimoto similarity.** Let $S^g$ be the set of bits that are set to 1 in the fingerprint of a generated molecule, and $S^t$ be the set of bits that are set to 1 in the target structure. The Tanimoto similarity is defined as $|S^g \cap S^t|/|S^g \cup S^t|$, where $|\cdot|$ denotes the number of elements in a set.

**Experimental details on QM9.** For the backbone of the noise prediction network, we use a three-layer MLP with 768, 512 and 192 hidden nodes as embedding to encode the fingerprint and add the output of it with the embedding of atom features $h$, which is then fed into the following EGNN. The EGNN has 256 hidden features and 9 layers. We train it 1500 epoch with a batch size of 64, a learning rate of 0.0001 with the Adam optimizer and an exponential moving average (EMA) with a rate of 0.9999. The energy function is trained with 1750 epoch with a batch size of 128, a learning rate of 0.0001 with the Adam optimizer and an exponential moving average (EMA) with a rate of 0.9999. Its EGNN has 192 hidden features and 7 layers. For the baseline cG-SchNet, we reproduce it using the public code. Since the default data split in cG-SchNet is different with ours, we train the cG-SchNet under the same data split with ours for a fair comparison. We report results at 200 epochs (there is no gain on similarity metric after 150 epochs). We evaluate the Tanimoto similarity on the whole test set.

**Experimental details on GEOM-Drug.** We use the same data split of GEOM-Drug with Hoogeboom et al. (2022), where training, validation and test set include 554K, 70K and 70K samples respectively. We train the noise prediction network and the energy function on the training set. For the noise prediction network, we use the recommended hyperparameters of EDM (Hoogeboom et al., 2022), where the EGNN has 256 hidden features and 4 layers, and the other part is the same as that in QM9. We train 10 epoch with a batch size of 64, a learning rate of 0.0001 with the Adam optimizer and an exponential moving average (EMA) with a rate of 0.9999. The backbone of the energy function is the same as that in QM9 and we train the energy function for 14 epoch. We evaluate the Tanimoto similarity on randomly selected 10K molecules from the test set.

## G ADDITIONAL RESULTS

### G.1 ABLATION STUDY ON NOISE PREDICTION NETWORKS AND ENERGY GUIDANCE

We perform an ablation study on the conditioning, i.e., use conditional or unconditional noise prediction networks, and the energy guidance. When neither the conditioning nor the energy guidance is adopted, a single unconditional model can't perform conditional generation, and therefore we only report its atom stability and the molecule stability. As shown in Table 5, Table 6 and Table 7, the conditional noise prediction network improves the MAE compared to the unconditional one, and the energy guidance improves the MAE compared to a sole conditional model. Both the conditioning and the energy guidance do not affect the atom stability and the molecule stability much.

Table 5: Effects of conditioning and energy guidance on a single quantum property $\mu$ (D).

| Conditional | Guidance | Scale | MAE↓ | Novelty↑ | AS↑ | MS↑ |
|---|---|---|---|---|---|---|
| × | × | - | - | 82.92 | 98.37 | 81.74 |
| × | ✓ | 0.1 | 1.415 | 83.60 | 98.33 | 81.16 |
| × | ✓ | 0.5 | 1.241 | 83.84 | 98.27 | 80.75 |
| ✓ | - | - | 1.138 | 84.04 | 98.14 | 80.04 |
| ✓ | ✓ | 0.1 | 1.071 | 84.36 | 98.18 | 80.08 |
| ✓ | ✓ | 0.5 | 0.935 | 84.06 | 98.20 | 80.05 |

Table 6: Effects of conditioning and energy guidance on a single quantum property $C_v$ ($\frac{\text{cal}}{\text{mol}}$K).

| Conditional | Guidance | Scale | MAE↓ | Novelty ↑ | AS↑ | MS↑ |
|---|---|---|---|---|---|---|
| ✗ | ✗ | - | - | 82.92 | 98.37 | 81.74 |
| ✗ | ✓ | 1 | 2.360 | 84.15 | 98.36 | 81.44 |
| ✗ | ✓ | 10 | 1.459 | 84.25 | 98.07 | 79.06 |
| ✓ | ✗ | - | 1.066 | 83.64 | 98.25 | 80.50 |
| ✓ | ✓ | 1 | 1.038 | 83.71 | 98.21 | 80.61 |
| ✓ | ✓ | 10 | 0.939 | 83.92 | 98.06 | 79.21 |

Table 7: Effects of conditioning and energy guidance on multiple quantum properties $C_v$, $\mu$.

| Conditional | Guidance | Scale | MAE1↓ | MAE2↓ | Novelty↑ | AS↑ | MS↑ |
|---|---|---|---|---|---|---|---|
| ✗ | ✗ | - | - | - | 82.92 | 98.37 | 81.74 |
| ✗ | ✓ | 1,0.1 | 2.381 | 1.421 | 83.81 | 98.31 | 80.52 |
| ✗ | ✓ | 10,1 | 1.501 | 1.157 | 84.11 | 98.02 | 78.20 |
| ✓ | ✗ | - | 1.075 | 1.163 | 85.45 | 97.99 | 77.02 |
| ✓ | ✓ | 1,0.1 | 1.053 | 1.097 | 85.20 | 97.96 | 77.08 |
| ✓ | ✓ | 10,1 | 0.982 | 0.916 | 85.51 | 97.58 | 73.94 |

## G.2 RESULTS ON NOVELTY, ATOM STABILITY AND MOLECULE STABILITY

For completeness, we also report novelty, atom stability and molecule stability on 10K generated molecules. Below we briefly introduce these metrics.

- Novelty (Simonovsky & Komodakis, 2018) is the proportion of generated molecules that do not appear in the training set. Specifically, let $G$ be the set of generated molecules, the novelty is calculated as $1 - \frac{|G \cap D_b|}{|G|}$. Note that the novelty is evaluated on $D_b$, since the reproduced conditional EDM and our method are trained on $D_b$. This leads to an inflated value compared to the one evaluated on the whole dataset (Hoogeboom et al., 2022).

- Atom stability (AS) (Hoogeboom et al., 2022) is the proportion of atoms that have the right valency. Molecule stability (MS) (Hoogeboom et al., 2022) is the proportion of generated molecules where all atoms are stable. We use the official implementation for the two metrics from the EDM paper (Hoogeboom et al., 2022).

As shown in Table 8 and Table 9, conditional EDM and EEGSDE with a small scaling factor have a slightly better stability, and EEGSDE with a large scaling factor has a slightly better novelty in general. The additional energy changes the distribution of generated molecules, which improves the novelty of generated molecules in general. Since there is a tradeoff between novelty and stability (see the caption of Table 5 in the EDM paper (Hoogeboom et al., 2022)), a slight decrease on the stability is possible when the scaling factor is large.

Table 8: Additional results on the novelty, the atom stability (AS) and the molecule stability (MS) of generated molecules targeted to a single quantum property.

| Method | Novelty↑ | AS↑ | MS↑ | Method | Novelty↑ | AS↑ | MS↑ |
|---|---|---|---|---|---|---|---|
| | $C_v$ | | | | $\mu$ | | |
| Conditional EDM | 83.64±0.30 | **98.25**±0.02 | 80.82±0.32 | Conditional EDM | 83.93±0.11 | 98.17±0.04 | **80.25**±0.40 |
| EEGSDE ($s$=1) | 83.53±0.18 | **98.25**±0.06 | **80.83**±0.33 | EEGSDE ($s$=0.5) | 83.85±0.20 | **98.18**±0.02 | **80.25**±0.18 |
| EEGSDE ($s$=5) | 83.57±0.17 | 98.16±0.04 | 80.22±0.34 | EEGSDE ($s$=1) | 84.43±0.21 | 98.17±0.04 | 80.06±0.35 |
| EEGSDE ($s$=10) | **83.78**±0.49 | 98.03±0.04 | 79.07±0.24 | EEGSDE ($s$=2) | **84.62**±0.31 | 98.06±0.02 | 78.92±0.30 |
| | $\Delta\varepsilon$ | | | | $\varepsilon_{\text{HOMO}}$ | | |
| Conditional EDM | 83.93±0.45 | **98.30**±0.04 | **81.95**±0.27 | Conditional EDM | 84.35±0.31 | 98.17±0.07 | 79.61±0.32 |
| EEGSDE ($s$=0.5) | 84.09±0.27 | 98.18±0.06 | 80.99±0.29 | EEGSDE ($s$=0.1) | 84.44±0.33 | **98.19**±0.03 | **79.81**±0.20 |
| EEGSDE ($s$=1) | 83.91±0.38 | 98.08±0.04 | 79.85±0.29 | EEGSDE ($s$=0.5) | 84.57±0.28 | 98.13±0.02 | 79.40±0.32 |
| EEGSDE ($s$=3) | **85.17**±0.69 | 97.76±0.03 | 77.43±0.37 | EEGSDE ($s$=1) | **84.77**±0.25 | 98.08±0.02 | 78.90±0.23 |
| | $\alpha$ | | | | $\varepsilon_{\text{LUMO}}$ | | |
| Conditional EDM | **84.56**±0.47 | 98.13±0.04 | 79.33±0.30 | Conditional EDM | 84.62±0.28 | 98.26±0.04 | **81.34**±0.29 |
| EEGSDE ($s$=0.5) | 84.35±0.31 | 98.19±0.03 | 80.08±0.34 | EEGSDE ($s$=0.5) | 84.37±0.31 | 98.25±0.04 | 81.18±0.31 |
| EEGSDE ($s$=1) | 84.45±0.33 | 98.17±0.04 | 80.04±0.36 | EEGSDE ($s$=1) | 84.70±0.34 | **98.27**±0.05 | 81.23±0.75 |
| EEGSDE ($s$=3) | 84.19±0.32 | **98.26**±0.03 | **80.95**±0.35 | EEGSDE ($s$=3) | **84.83**±0.30 | 98.14±0.01 | 80.00±0.21 |

Table 9: Additional results on the novelty, the atom stability (AS) and the molecule stability (MS) of generated molecules targeted to multiple quantum properties.

| Method | Novelty↑ | AS↑ | MS↑ |
|---|---|---|---|
| | $C_v,\ \mu$ | | |
| Conditional EDM | 85.31±0.43 | **98.00**±0.07 | **77.42**±0.80 |
| EEGSDE ($s_1$=10, $s_2$=1) | **85.62**±0.86 | 97.67±0.08 | 74.56±0.54 |
| | $\Delta\varepsilon,\ \mu$ | | |
| Conditional EDM | 85.06±0.27 | **97.96**±0.00 | **75.95**±0.30 |
| EEGSDE ($s_1$=$s_2$=1) | **85.56**±0.56 | 97.61±0.04 | 72.72±0.27 |
| | $\alpha,\ \mu$ | | |
| Conditional EDM | 85.18±0.35 | **98.00**±0.06 | **77.96**±0.33 |
| EEGSDE ($s_1$=$s_2$=1.5) | **85.36**±0.03 | 97.99±0.06 | 77.77±0.26 |

### G.3 VISUALIZATION OF THE EFFECT OF THE SCALING FACTOR

We visualize the effect of the scaling factor in Figure 4, where the generated structures align better as the scaling factor grows.

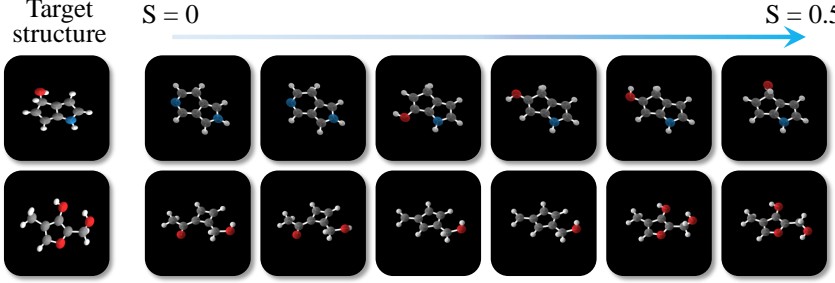

Figure 4: Visualization of the effect of the scaling factor on QM9. As the scaling factor grows, the generated structures align better with the target structure. $S = 0$ corresponds to the conditional EDM.

### G.4 REPRODUCE

We compare our reproduced results and the original results of conditional EDM (Hoogeboom et al., 2022) in Table 10. The results are consistent.

Table 10: The reproduced results of conditional EDM and L-bound are consistent with the original ones.

| Quantum property
Unit | $\alpha$
Bohr$^3$ | $\Delta\varepsilon$
meV | $\varepsilon_{\text{HOMO}}$
meV | $\varepsilon_{\text{LUMO}}$
meV | $\mu$
D | $C_v$
$\frac{\text{cal}}{\text{mol}}$K |
|---|---|---|---|---|---|---|
| Conditional EDM (Hoogeboom et al., 2022) | 2.76 | 655 | 356 | 584 | 1.111 | 1.101 |
| Conditional EDM (reproduce) | 2.79 | 674 | 371 | 593 | 1.118 | 1.054 |
| L-bound (Hoogeboom et al., 2022) | 0.10 | 64 | 39 | 36 | 0.043 | 0.040 |
| L-bound (reproduce) | 0.09 | 65 | 39 | 36 | 0.043 | 0.040 |

### G.5 GENERATING MOLECULES WITH TARGET STRUCTURES ON GEOM-DRUG

We plot generated molecules on GEOM-Drug in Figure 5, and it can be observed that the atom types of generated molecules with EEGSDE often match the target better than conditional EDM.

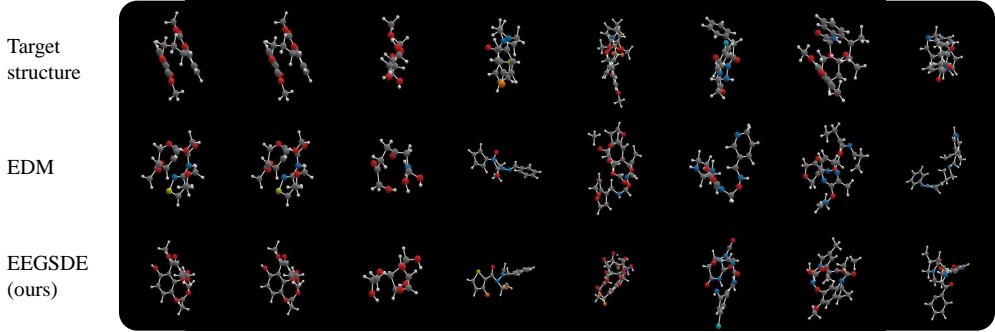

Figure 5: Generated molecules on GEOM-Drug.

### G.6 THE DISTRIBUTIONS OF PROPERTIES

We plot the distribution of the properties of the training set, as well as the distribution of the properties of generated molecules (calculated by the Gaussian software). As shown in Figure 6, these distributions match well.

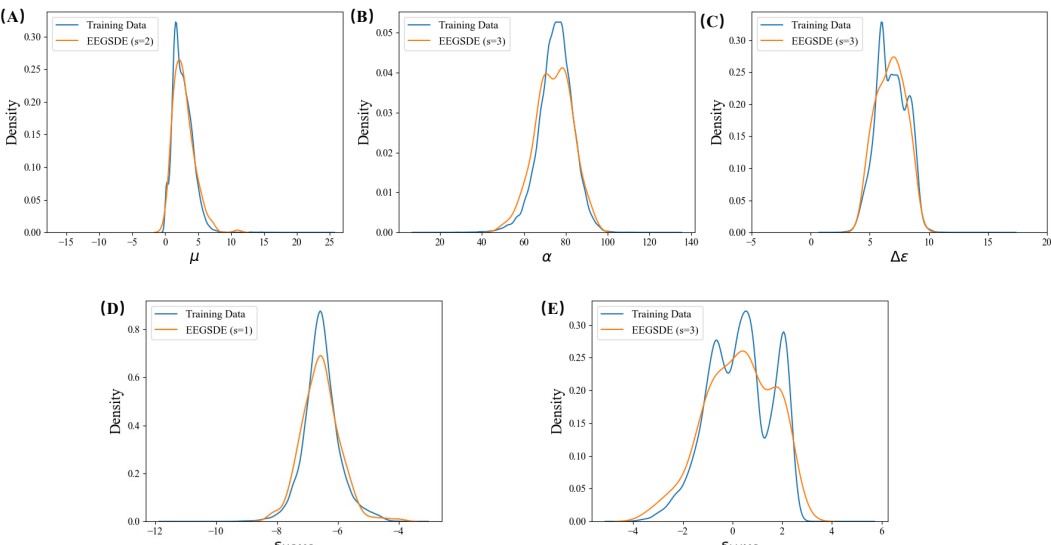

Figure 6: The distribution of the properties of the training set vs. the distribution of the properties of molecules generated by EEGSDE.

