# OpenReview forum: "Equivariant Energy-Guided SDE for Inverse Molecular Design"
_ICLR.cc/2023/Conference — ICLR 2023 poster_

### Official Review · Reviewer_1ipW · 2022-10-14

**Confidence:** 4
**Correctness:** 4
**Technical Novelty And Significance:** 4
**Empirical Novelty And Significance:** 2
**Recommendation:** 8

**Clarity, Quality, Novelty And Reproducibility:**

The paper is clearly written and provides a thorough related work section to help understand the main contribution of the paper (inverse molecular design).

Novelty-wise this work is somewhat incremental, as it builds on a lot of recent work (e.g. SDE formulation of denoising diffusion models, EGNNs) although to the best of my knowledge it is the first to propose how to do inverse molecular design with these.

The authors provide code in the supplementary information to reproduce some of the experiments, although the quality of the code could significantly be improved.

**Strength And Weaknesses:**

Overall, the paper is somewhat incremetal but in general technically sound and a pleasure to read.

Strengths:
* Clear motivation to solve a real problem (inverse molecular design)
* A lot of effort went into explaining previous methods. This is a fantastic introductory paper that requires no reference reading for context.
* Simple and elegant formulation
* Code provided in the supplementary material

Weaknesses:
* Main results on Tables 1, 2, 3 could be provided with standard deviations to assess how stable results are depending on the chosen seed
* Supplied code could be cleaned / improved with clearer instructions on how to run the experiments in the paper.

Other comments:

* It would be fantastic if the authors physically motivated the choice for using energy functions (Eq. 6). The reverse SDE is presented without a lot of justification
* In Section 6 the authors claim that they tried a binary cross-entropy loss for generating molecules with desired structures, but ended up using a mean squared error loss. In Appendix D, however, the binary cross entropy is the one described for this task.
* It is somewhat unclear to me whether the time-dependent property prediction model is jointly trained with the score model or whether this is trained separately. It would be great if the authors could clarify along these lines.

**Summary Of The Paper:**

The authors present an energy-guided score model formulation for the task of equivariantly generating molecules. This work builds upon lots of recent references on SDE score matching models, equivariant networks (EGNNs) and specifically the work of Hoogeboom et al (2022), EDM. The main contribution of the paper is the addition of an energy function in the score matching algorithm, which can be used to generate molecules with desired properties.

**Summary Of The Review:**

Overall, this is a good contribution to ICLR. Although the approach is marginally incremental, I believe the work is novel and shows how to tackle a fundamental problem in atomistic ML, how perform inverse molecular design using denoising diffusion models on a regression setting.

I would recommend acceptance upon revision of several minor issues.

---

> ### Author Response · Authors · 2022-11-15
> **Author Response to Reviewer 1ipW**
>
> Thanks for the acknowledgement of our contributions and the valuable comments.
>
> ## Response to Q1.
>
> *``Main results on Tables 1, 2, 3 could be provided with standard deviations to assess how stable results are depending on the chosen seed''*
>
> Thanks for the suggestions. As suggested, we repeat all experiments three times by change the random seed and report the mean and standard deviations in the revised version (see Table 1, Table 2 and Table 3). The results are still consistent with that in the original version. We also conduct two sample t-test between the results of EEGSDE and EDM. The results show our EEGSDE outperform EDM significantly with the significance value $p<0.01$.
>
> ## Response to Q2.
>
> *``Supplied code could be cleaned / improved with clearer instructions on how to run the experiments in the paper.''*
>
> Thanks for the suggestions. We improve the code instruction in the revised supplementary material.
>
> ## Response to Q3.
>
> *``It would be fantastic if the authors physically motivated the choice for using energy functions (Eq. 6). The reverse SDE is presented without a lot of justification''*
>
> Thanks for the suggestion.
> The choice of the energy function depends on the application. For example, we design the energy function according to consistency between the molecule $\pmb{z}$ and the property $c$, where a low energy represents a well consistency. As the generation process in Eq.(6) proceeds, the gradient of the energy function encourages generated molecules to have a low energy, and consequently a well consistency. In this case, we can expect the generated molecule $\pmb{z}$ aligns well with the property $c$. We add this discussion in Section 4.4 in the revised version.
>
> Generally, the energy function changes the distribution of generated molecules. The new distribution can be approximately formalized as a product of experts [1], i.e., the product of the energy-based model $\exp(-E(\pmb{z}, c, 0))$ at time $t$=0 and the original molecular distribution defined by the SDE without energy guidance.
>
>
>
> ## Response to Q4.
>
> *``In Section 6 the authors claim that they tried a binary cross-entropy loss for generating molecules with desired structures, but ended up using a mean squared error loss. In Appendix D, however, the binary cross entropy is the one described for this task.''*
>
> Thanks for the comment. The choice of the energy function is flexible and can be different to the training loss of the fingerprint prediction model. In initial experiments, we also try binary cross entropy loss for the energy function, but we find it causes the generation process unstable. We make this more clear in the second paragraph of Section 6 in the revised version.
>
> ## Response to Q5.
>
>
> *``It is somewhat unclear to me whether the time-dependent property prediction model is jointly trained with the score model or whether this is trained separately. It would be great if the authors could clarify along these lines.''*
>
> Thanks for the comment. The score model and the time-dependent property prediction model are trained separately. We make it clear in Section 5.1 in the revised version.
>
> [1] Zhao et al., EGSDE: Unpaired Image-to-Image Translation via Energy-Guided Stochastic Differential Equations

---

> > ### Comment · Reviewer_1ipW · 2022-11-16
> > **Acknowledgement of revision**
> >
> > The authors have successfully addressed my comments in this revision.

---

> > > ### Author Response · Authors · 2022-11-17
> > > **Thanks**
> > >
> > > Great! Thank you again for your efforts and acknowledgment of our work.

---

### Official Review · Reviewer_ZV34 · 2022-10-22

**Confidence:** 3
**Correctness:** 2
**Technical Novelty And Significance:** 2
**Empirical Novelty And Significance:** Not applicable
**Recommendation:** 6

**Clarity, Quality, Novelty And Reproducibility:**

* The paper is easy to read and well organized.
* The energy function is novel, while the other parts appear to be adaptations of existing ones.



**Strength And Weaknesses:**

[Strength]
* Incorporating energy functions is well supported by the theories.
* In the task of generating molecules with desired quantum properties, the proposed framework significantly improves its baseline, EDM.
* The paper is easy to read and well organized.
* The visualizations help to understand how the model behaves in different settings.


[Weaknesses]
* Although their approach is novel, the core model relies on the existing model EDM with just adding energy functions.
* EDM seems to be generating more stable molecules. Insights behind this would be helpful.
* Why not compare the model by Gebauer et al. 2022 in the task of generating molecules with target structures? This task seems to be motivated by that paper.
* The authors, however, include EDM as a baseline. How did you create a molecule with target structures using EDM? This comparison appears to be unfair because there are no explicit terms for target structures in EDM.


**Summary Of The Paper:**

This paper proposes a new diffusion model for generating 3D molecules using the guidance energy function. In order to incorporate various energy functions into the proposed framework, the authors theoretically demonstrate that those functions are invariant to transformations, which is an essential feature for 3D molecule representations. The proposed model outperforms its predecessor by a wide margin. Additionally, the representation makes it easier to comprehend how the model performs as intended.

**Summary Of The Review:**

Despite the fact that the novel part is minor, it is theoretically supported, which is a plus. However, generating molecules with target structures appears to be unfair and requires improvement.

---

> ### Author Response · Authors · 2022-11-15
> **Author Response to Reviewer ZV34**
>
> Thanks for the valuable comments. Below we address concerns of Reviewer ZV34, and we are glad to reply if further questions are asked.
>
>
> ## Response to Q1.
>
>
> *``Although their approach is novel, the core model relies on the existing model EDM with just adding energy functions.''*
>
>
> Thanks for the comment. Our contribution is to use a principled method to solve an important real problem, i.e., improving the alignment between generated molecules and condition for inverse molecular design.
>
>
>
> ## Response to Q2.
>
> *``EDM seems to be generating more stable molecules. Insights behind this would be helpful.''*
>
>
> Thanks for the comment. The additional energy changes the distribution of generated molecules, which improves the novelty of generated molecules in general. Since there is a tradeoff between novelty and stability (see the caption of Table 5 in the EDM paper [1]), a slight decrease on the stability is possible. We add this discussion in Appendix H.2 in the revised version.
>
> We argue that the benefit brought by the energy function outweighs the slight decrease of the stability, since the alignment between generated molecules and conditions is critical in inverse molecular design while we can easily screen unstable molecules in practice. Overall, the improvement on the MAE metric is significant, which compensates the slight decrease of the stability.
>
>
>
>
> ## Response to Q3.
>
> *``Why not compare the model by Gebauer et al. 2022 in the task of generating molecules with target structures? This task seems to be motivated by that paper.''*
>
>
> Thanks for the comments. As suggested, we compare the model by Gebauer et al. 2022 (cG-SchNet) in the task of generating molecules with target structures. EEGSDE outperform cG-SchNet significantly (0.750 vs 0.499 in similarity). All experiments is repeated three times to eliminate the influence of randomness. We add this experiment in Table 3 in the revised version, and present experimental details of cG-SchNet in Appendix G.4 in the revised version.
>
> ## Response to Q4.
>
> *``The authors, however, include EDM as a baseline. How did you create a molecule with target structures using EDM? This comparison appears to be unfair because there are no explicit terms for target structures in EDM.''*
>
> Thanks for the comment. Indeed, we use a conditional EDM, which takes the target structure as an input. Specifically, it feeds the the target structure into the network by encoding its fingerprint using a three-layer MLP. The only difference between EDM and EEGSDE is that EEGSDE uses an additional energy function to strengthen the condition signal (other parts including the conditional noise prediction network are the same). Therefore, the comparison is fair.
>
> We are glad to reply if there is any further question.
>
>
> [1] Hoogeboom et al., Equivariant Diffusion for Molecule Generation in 3D

---

> > ### Author Response · Authors · 2022-11-29
> > **Sincerely looking forward to the further discussions**
> >
> > Dear reviewer ZV34,
> >
> > We are wondering if our response and revision have resolved your concerns. If our response has addressed your concerns, we would highly appreciate it if you could re-evaluate our work and consider raising the score.
> >
> > If you have any additional questions or suggestions, we would be happy to have further discussions.
> >
> > Best regards,
> >
> > The Authors

---

> > > ### Comment · Reviewer_ZV34 · 2022-11-30
> > > **Additional question**
> > >
> > > Thank you for your detailed responses, which have helped me to clarify a number of issues. I have a further question. The major novel part is the energy function, however, it only improves MAE errors while not affecting other metrics (sometimes even lowering MS or AS) according to Tables 4,5, and 6 in Appendix. With this observation, can we still conclude that the energy function can be regarded as a crucial part?

---

> > > > ### Author Response · Authors · 2022-12-01
> > > > **Response to the additional question**
> > > >
> > > > Thanks for the further comment.
> > > >
> > > > We argue that the energy function is a crucial part, since the significant improvement on MAE is sufficient to verify its effectiveness for **inverse molecular design**, which is the focus of this paper.
> > > >
> > > > Note that MAE directly measures the performance on inverse molecular design, i.e., how generated molecules align with the condition, and therefore MAE is the main metric of this paper.
> > > >
> > > > As for novelty, AS and MS metrics, we include them for completeness following the unconditional molecule generation experiment in EDM paper [1], although they do not measure the performance on inverse molecular design.

---

> ### Author Response · Authors · 2022-12-07
> **A new experimental update**
>
> Dear reviewer ZV34,
>
> Thank you again for the valuable and constructive comments. We update an experiment that uses the **Gaussian software** to calculate MAE. In this case, our EEGSDE still has a better MAE than the EDM baseline. The Gaussian software calculates properties according to theories of quantum chemistry, which provides new evidence for the effectiveness of our method. We hope this update could serve as a reason to further increase the score.

---

### Official Review · Reviewer_ChXk · 2022-10-23

**Confidence:** 4
**Correctness:** 3
**Technical Novelty And Significance:** 2
**Empirical Novelty And Significance:** 2
**Recommendation:** 6

**Clarity, Quality, Novelty And Reproducibility:**

**Clarity**

Clarity in the formulation is detailed and well-written. I think clarity in the results can be improved as indicated in the above sections.

**Quality**

Quality of the presented experiments appears to done well. The general results sections could be improved by adding clearer conclusions based on the experiments.

**Novelty**

While the authors introduce a novel equivariant diffusion formulation, it does appear to be a bit incremental from the work performed by EDM which is the primary baseline the authors compare to.

**Reproducibility**

The authors provide a good amount of detail in their detail and appendix along with a reproducibility statement.

**Strength And Weaknesses:**

**Strengths**

* The paper introduces a new equivariant molecular diffusion method and describes the different pieces, including various definitions and formulations, in great detail.
* The results presented in the experiments seem to indicate some advantages of using EEG-SDE compared to EDM.

**Weaknesses**

* Some of the figures, especially Figure 1 and Figure 4, could benefit from greater detail to the overall message they are trying to convey. In Figure 1, the different images could have additional labels to guide the reader as to what the different sections represent. In Figure 4, it is unclear what the authors are trying to show with the caption not being clear about the purpose and it being hard accompanying text being hard to find.
* The results on the GEOM-Drug dataset are only briefly mentioned in the text. It would have been nice to see them included in more cohesive manner, potentially in one of the prior tables.
* The overall message from the results is somewhat unclear. It would be nice if the paper could summarize some key takeways for EEGSDE based on the results of the experiments?

**Additional Questions**

* Fitting to properties appears to be applicable to one property at a time given the scalar signal. How does the multi-property case shown in Figure 4 work in that instance? Is it mainly a loss over a vector? If so, how is it implemented?
* Did you consider trying other equivariant models (SE3-Transformer, DimeNet, etc)? Why, why not?
* It would good to have the definitions for the metrics in Table 1 and Table 2 (Similarity, Novelty, Atomic Stability (AS), Molecular Stability (MS)). Putting them in the appendix is fine.
* What is the distinction between EDM and this method? A clear distinction would provide more clarity on the novelty - from what I can tell it is the inclusion of SDE formulation. As such, it would be good to clarify the difference between SDE and diffusion models in general terms first. Section 4.1 has SDE definitions, but not a contrast to standard diffusion.
* How would you compare your work to Torsion Diffusion (Jing, Bowen, et al. "Torsional Diffusion for Molecular Conformer Generation." arXiv preprint arXiv:2206.01729 (2022).)?
* Could you provide more details on the molecular fingerprint - how does it capture 3D similarity related to positions?

**Summary Of The Paper:**

The paper proposes a new method for 3D molecular generation called energy-guided stochastic differential equations (EEGSDE), which introduces an equivariant diffusion model based on SDE formulations. The authors first introduce the problem setting of 3D molecular design, as well as past work in molecular design and related work including diffusion models and guidance of diffusion models. Subsequently, the paper introduces relevant background and formulations (3D molecule representation, equivariance and Center of Mass modeling) and outlines the mathematical details of EEG-SDE, including the SDE formulation and an extension for an equivariant SDE formulation based on an equivariant neural network backbone (EGNN). After outlining the entire algorithmic flow with all relevant and priorly introduced components, the authors introduce a set of molecular design experiments based primarily on the QM9 dataset.

In their experiments, the authors first train EEG-SDE on the QM9 dataset and compare it's performance to equivariant diffusion models (EDM) along a couple of metrics (MAE, Novelty, Atomic Stability, Molecular Stability). In their formulation, the authors also include a scaling factor, $s$, which serves to modulate the strength of a given loss signal and is varied in many of the authors' experiments. The first set of presented results show general competitiveness with EDM with outperformance along some metrics and lower performance on others. The authors then show an experiment generating molecules based on a target structure where the similarity of the generated molecule to the target structure is based on a molecular fingerprint. Based on the similarity metric EEG-SDE appears to outperform SDE.

**Summary Of The Review:**

While I appreciated the thoroughness of the description of the method and all its different parts, in my view the reasons to reject outweigh the reasons to accept the paper in its current form. The novelty appears a bit incremental compared to EDM and the presentation of the methods and results is a bit unclear to be able to see clear differentiation.

------
Score revised from 5 -> 6 based on discussion and adjustments made during author response.

---

> ### Author Response · Authors · 2022-11-15
> **Author Response to Reviewer ChXk (1/2)**
>
> Thanks for the valuable suggestions. Below we address concerns of Reviewer ChXk, and we are glad to reply if further questions are asked.
>
> ## Response to Q1.
>
> *``Some of the figures, especially Figure 1 and Figure 4, could benefit from greater detail to the overall message they are trying to convey. In Figure 1, the different images could have additional labels to guide the reader as to what the different sections represent. In Figure 4, it is unclear what the authors are trying to show with the caption not being clear about the purpose and it being hard accompanying text being hard to find.''*
>
> Thanks for the suggestion. We improve Figure 1 and Figure 4 as suggested in the revised version (Figure 4 corresponds to Figure 3 in the revised version). Figure 4 visualizes the effect of two scaling factors when targeting to both a structure and a quantum property, which demonstrates that our EEGSDE can generate molecules with multiple target properties.
>
> ## Response to Q2.
>
> *``The results on the GEOM-Drug dataset are only briefly mentioned in the text. It would have been nice to see them included in more cohesive manner, potentially in one of the prior tables.''*
>
> Thanks for the suggestion. As suggested, we move the results of GEOM-Drug from the Appendix to Table 3 in the main body in the revised version.
>
> ## Response to Q3.
>
> *``The overall message from the results is somewhat unclear. It would be nice if the paper could summarize some key takeways for EEGSDE based on the results of the experiments?''*
>
> Thanks for the suggestion.  The first key takeaway is that molecules generated by our EEGSDE align better with the condition (e.g., desired quantum properties and target structures) than molecules generated by the EDM baseline, which  are verified by Table 1, 2, 3 and Figure 2. This indicates that EEGSDE generates more promising candidates for applications such as the virtual screening. The second key takeaway is that our EEGSDE is able to generate molecules targeted to multiple conditions, which are verified by Table 2 and Figure 3. We highlight these takeaways in the conclusion paragraph in Section 5.2 \& 6.2 in the revised version. These key takeaways are as expected with the guidance of the energy function, which is the key distinction discussed in Q7.
>
> ## Response to Q4.
>
>
> *``Fitting to properties appears to be applicable to one property at a time given the scalar signal. How does the multi-property case shown in Figure 4 work in that instance? Is it mainly a loss over a vector? If so, how is it implemented?''*
>
> Thanks for the comment. In short, we fit to multi-property by adding energies on single properties (mentioned in the last but one paragraph of Section 5 in the original version), and then using the gradient of the summed energy to guide the reverse SDE as described in Eq.(6). Specifically, let $\pmb{c} = (c_1, c_2, \dots, c_K)$ be $K$ different properties, and let $E_k(\pmb{z}_t, c_k, t) = s_k |g_k(\pmb{z}_t, t) - c_k|^2$ be the energy function for the $k$-th property, where $g_k(\pmb{z}_t, t)$ is a time-dependent property prediction network of the $k$-th property. The energy function for the multi-property case is defined as $E(\pmb{z}_t, \pmb{c}, t) = \sum_k E_k(\pmb{z}_t, c_k, t)$, and we use its gradient to guide the reverse SDE as described in Eq.(6). We add these details in the second paragraph in Section 5 in the revised version.
>
>
>
> ## Response to Q5.
>
> *``Did you consider trying other equivariant models (SE3-Transformer, DimeNet, etc)? Why, why not?''*
>
>
> Thanks for the comment. For the equivariant model used in diffusion model, we directly use the same backbone with EDM for a fair comparison. As for the equivariant model used in the energy function, we notice that the EGNN (i.e., the one used in this paper) outperforms a large family of equivariant models including SE3-Transformer and DimeNet for the molecular property prediction on the QM9 dataset (see Table 3 in EGNN paper [1]). Therefore, we choose EGNN as a backbone for the prediction network.
>
>
>
> ## Response to Q6.
>
> *``It would good to have the definitions for the metrics in Table 1 and Table 2 (Similarity, Novelty, Atomic Stability (AS), Molecular Stability (MS)). Putting them in the appendix is fine.''*
>
> Thanks for the suggestion. We put definitions for Novelty, Atomic Stability and Molecular Stability in Appendix H.2 in the revised version.

---

> > ### Author Response · Authors · 2022-11-15
> > **Author Response to Reviewer ChXk (2/2)**
> >
> > ## Response to Q7.
> >
> > *``What is the distinction between EDM and this method? A clear distinction would provide more clarity on the novelty - from what I can tell it is the inclusion of SDE formulation. As such, it would be good to clarify the difference between SDE and diffusion models in general terms first. Section 4.1 has SDE definitions, but not a contrast to standard diffusion.''*
> >
> > Thanks for the comment. Indeed, the main distinction is that our method incorporates energy functions to guide the molecular generation, which are not employed in EDM. The energy function changes the distribution of generated molecules and strengthens the control from the condition signal.
> >
> > Theoretically, since incorporating a naive energy function is likely to destroy the equivariance of the generation process, we prove the energy function should be invariant to orthogonal transformations to ensure the equivariance.
> >
> > Empirically, this additional energy term encourages our method to generate samples that consistently and significantly align better with the condition than EDM (see the MAE metric in Table 1-2, the similarity metric in Table 3, and Figure 2).
> >
> > As for the SDE formulation, we include it since it is a principled and general framework [2] that covers discrete time diffusion model as a specific type of discretization.
> >
> >
> > ## Response to Q8.
> >
> > *``How would you compare your work to Torsion Diffusion (Jing, Bowen, et al. "Torsional Diffusion for Molecular Conformer Generation." arXiv preprint arXiv:2206.01729 (2022).)?''*
> >
> > Thanks for the suggestion. Torsion Diffusion focuses on a different task, i.e., the conformer generation. Thus, experimental results presented in Torsion Diffusion are not comparable.
> > Technically, our method is also applicable to Torsion Diffusion by incorporating an energy function that encourages the alignment between the 2D molecular graph and torsion angles.
> >
> >
> >
> > ## Response to Q9.
> >
> > *``Could you provide more details on the molecular fingerprint - how does it capture 3D similarity related to positions?''*
> >
> > Thanks for the suggestion. A molecular fingerprint is a bitmap $c=(c_1,\dots, c_L)$ that captures 3D information of a molecule by recording the presence or absence of substructures in the molecule, and the 3D similarity between two molecules is captured by comparing their corresponding bitmaps. Specifically, a substructure is mapped to a specific position $l$ in the bitmap, and the corresponding bit $c_l$ will be 1 if the substructure exists in the molecule and will be 0 otherwise.
> >
> > Then, the similarity between two molecules can be captured by comparing their corresponding bitmaps. Specifically, let $S^g$, $S^t$ be the sets of bits that are set to 1 in the fingerprints of two molecules respectively. The similarity is computed as $|S^g \cap S^t| / |S^g \cup S^t|$, where $|\cdot|$ denotes the number of elements in a set.
> >
> > We add these details in the first paragraph in Section 6 and the second paragraph in Section 6.1 in the revised version.
> >
> > [1] Satorras et al., E(n) equivariant graph neural networks.
> >
> > [2] Song et al., Score-Based Generative Modeling through Stochastic Differential Equations

---

> > > ### Comment · Reviewer_ChXk · 2022-11-16
> > > **Further Questions on Q9**
> > >
> > > Thanks for providing additional details. In regards to Q9, I was referring to which particular kind of molecular fingerprint you are using and why you made that choice. There are a couple of established options in the literature.

---

> > > > ### Author Response · Authors · 2022-11-16
> > > > **Response to Further Questions on Q9**
> > > >
> > > > Thanks for the timely reply!
> > > > We use the path-based fingerprint FP2 (provided by the "Open Babel" library). We use it following cG-SchNet [3], which is a prior work on generating molecules conditioned on fingerprint. We add cG-SchNet as a baseline in Table 3 in the revised version.
> > > >
> > > > [3] Gebauer et al., Inverse design of 3d molecular structures with conditional generative neural networks.

---

> > ### Comment · Reviewer_ChXk · 2022-11-16
> > **Further Questions on Q5**
> >
> > Thanks for the additional details - could you provide more thoughts/intuition on how you would expect other equivariant models to perform in your setup? Would you generally expect similar performance or are there particular things about the EGNN architecture that make it appealing other than empirical results?

---

> > > ### Author Response · Authors · 2022-11-16
> > > **Response to Further Questions on Q5**
> > >
> > > Thanks for the timely reply!
> > > We expect other equivariant models (e.g., SE3-Transformer and DimeNet) also perform well, and they will have a similar performance.
> > > Specifically, as long as an architecture performs well in the task of property prediction, it is applicable to our guidance model.
> > > We add this discussion and cite other possible equivariant models for our method including SE3-Transformer and DimeNet in the revised version.

---

> ### Author Response · Authors · 2022-12-07
> **A new experimental update**
>
> Dear reviewer ChXk,
>
> Thank you again for the great efforts and the valuable comments. We update an experiment that uses the **Gaussian software** to calculate MAE. In this case, our EEGSDE still has a better MAE than the EDM baseline. The Gaussian software calculates properties according to theories of quantum chemistry, which provides new evidence for the effectiveness of our method. We hope this update could serve as a reason to further increase the score.

---

### Official Review · Reviewer_BGNw · 2022-10-31

**Confidence:** 4
**Correctness:** 3
**Technical Novelty And Significance:** 3
**Empirical Novelty And Significance:** 3
**Recommendation:** 6

**Clarity, Quality, Novelty And Reproducibility:**

Clarity: The paper is well-structured and very readable
Quality: The paper is of high quality in the sense of how the algorithms, tables and figures are organised and presented.
Novelty: The authors primarily build their work on Equivariant Diffusion models. The introduction of the property prediction process as a guiding force seems however relatively novel. The extent to which this is an effective addition is however unclear at this moment in time.
Reproducibility: Clear training details are missing in the main body of the paper. Code is however made available.

**Strength And Weaknesses:**

With their proposed method the authors tackle the currently relevant and exciting problem of conditional molecular generation. The explicit focus on conditional generation is an important contribution to the molecular design space. The paper is very well structured and sufficiently places their contribution in the context of prior work on diffusion models. Although, this connection could be strengthened by making this connection explicit in the name of the method (eg. Equivariant Energy-Guided Diffusion Models).

However, there are a number of issues with the paper that I would like to see addressed:

1. Conceptually, how does the conditional generation using the prediction method differ from the conditional generation as done by Hoogeboom et al. Most importantly, what are the differences between conditional samples generated by the presented method and the EDM method?
2. I believe it would be interesting/needed to include a note as to why the loaded term “Energy Model” is used to refer to the property prediction model. Aside from the gradient of the property prediction network being used as a guiding force, I do not see any direct relation between the MSE loss and the molecular energy. Using the term “Energy” in this context might give the reader the impression some form of potential energy or free energy guidance is used.
3. It is unclear to me how representative the MAE as a metric for evaluation is. Am I correct in assuming that the trained property prediction model of the presented method is used to evaluate the others?
4. Related, it would be good if the authors could clarify how representative the current novelty scores are and which subsets of the dataset are used for calculating the novelty score for each model. As I understand it, only a subset of the data is used and as such the novelty score should be inflated compared to when using the entire dataset. Contrary to this expectation, the novelty score for the SDM method seems lower than presented in the original paper (Table 2 of Hoogeboom et al.)
5. Also related, I’m unsure about how representative atom stability and molecular stability are as a metric. Especially in the context of this work where the bonds have to be estimated heuristically.


**Summary Of The Paper:**

In this work, the authors present an extension of equivariant Diffusion Models (as presented by Hoogeboom et al.) for conditional molecular generation. The method, named “Equivariant Energy-Guided SDE” (EEGSDE), introduces an additional property prediction network that acts as a guiding force in the generation process. The property prediction network is trained in parallel with the reverse diffusion process, and the network's gradient is taken to be the additional force.

The authors evaluate their conditional generation on multiple metrics—generation novelty, atom stability, molecule stability and the mean absolute error given by the trained property classifier. Under these metrics, the main improvement of the work is in terms of MAE and novelty, with the note that it is unclear how representative the novelty metrics are due to the training procedure of the prediction network.

**Summary Of The Review:**

At this moment in time, I do not believe the paper to be ready for publication at the ICLR conference. Most importantly, the current questions regarding the metrics used for evaluating the method make it hard to judge the significance of the work. With these questions addressed in the rebuttal, I would be willing to increase my score.

---

Based on the newly presented results using Gaussian software to compute the properties of generate molecules I have raise my score (5->6).

---

> ### Author Response · Authors · 2022-11-15
> **Author Response to Reviewer BGNw (1/2)**
>
> Thanks for the valuable suggestions. We make training details more clear in the main body as suggested (see Section 5.1 \& 6.1 in revised version), and will rename our method Equivariant Energy-Guided Diffusion Models in the final version. Below, we address other concerns of Reviewer BGNw, and we are glad to reply if further questions are asked.
>
> ## Response to Q1.
>
> *``Conceptually, how does the conditional generation using the prediction method differ from the conditional
> generation as done by Hoogeboom et al.''*
>
> Conceptually, our method additionally incorporates an energy function compared to EDM, which changes the distribution of generated molecules. The new distribution can be approximately formalized as a product of experts [3], i.e., the product of the energy-based model $\exp(-E(\pmb{z}, c, 0))$ at time $t$=0 and the original molecular distribution defined by the SDE without energy guidance. As a result, the incorporation of an energy function strengthens the control from the condition signal.
>
>
> *``Most importantly, what are the difference between conditional samples generated by the presented method and the EDM method?''*
>
> Our method generates samples that align better with the condition than EDM. For example, Table 1 and Table 2 show quantum properties of molecules generated by our method are closer to the desired ones than EDM, as measured by the mean absolute error (e.g., 0.777 in our method vs 1.123 in EDM for the dipole moment $\mu$); Table 3 shows molecules generated by our method have a better similarity to the target structure than baselines (e.g., 0.750 in our method vs 0.671 in EDM).
>
> ## Response to Q2.
>
> *``I believe it would be interesting/needed to include a note as to why the loaded term “Energy Model” is used to
> refer to the property prediction model. Aside from the gradient of the property prediction network being used
> as a guiding force, I do not see any direct relation between the MSE loss and the molecular energy. Using the
> term “Energy” in this context might give the reader the impression some form of potential energy or free
> energy guidance is used.''*
>
> Thanks for the comment. Indeed, the term “energy” in this paper refers to a general notion in statistical machine learning, which is a scalar function that captures dependencies between input variables (see abstract in [1]). Thus, the “energy” in this paper can be set to a MSE loss when we want to capture how the molecule align with the property. Also, the “energy” in this paper does not exclude potential energy or free energy in chemistry, and they might be applicable when we want to generate molecules with small potential energy or free energy. We add this discussion in Section 4.4 in the revised version.
>
> ## Response to Q3.
> *``It is unclear to me how representative the MAE as a metric for evaluation is.''*
>
> Thanks for the comment. We use the MAE metric following the EDM [2], which is the most direct baseline of our method. Note that MAE is the main focus in this paper, since it measures how generated molecules align with the condition (e.g., desired quantum properties) in conditional generation.
>
> *``Am I correct in assuming that the trained property prediction model of the presented method is used to evaluate the others?''*
>
> The trained property prediction model of the presented method (denoted as the guidance model) *is not* used to evaluate the others. Indeed, we train an addition evaluation model, which is different to the guidance model from two aspects:
> * The evaluation model is trained on a different data split to that of the guidance model. Specifically, as mentioned in Lines 5-8 in the second paragraph in Section 5 (in the original version), we split the training set into two non-overlapping halves $D_a$ and $D_b$ equally following the EDM paper [2], where the guidance model is trained on $D_b$, and the evaluation model is trained on $D_a$.
> * The evaluation model has a different architecture to the guidance model, e.g. different number of layers and hidden features. In addition, the guidance model is time-dependent while the evaluation model is not.
>
> This ensures no information leak occurs when evaluating our EEGSDE, and therefore ensures the fairness.
> We make this experimental detail more clear in Section 5.1 in the revised version.

---

> > ### Author Response · Authors · 2022-11-15
> > **Author Response to Reviewer BGNw (2/2)**
> >
> > ## Response to Q4.
> > *``Related, it would be good if the authors could clarify how representative the current novelty scores are.''*
> >
> > Thanks for the comment. For fairness and completeness, we use the novelty metric following the EDM paper [2], although it is not our main focus.
> >
> > *``Which subsets of the dataset are used for calculating the novelty score for each model.''*
> >
> > The novelty scores for both EDM and our method are calculated on $D_b$, which is the same set used to train EDM and our method (as mentioned in the last three lines in the second paragraph in Section 5 in the original version). We make this experimental detail more clear in Appendix H.2 in the revised version.
> >
> > *``As I understand it, only a subset of the data is used and as such the novelty score should be inflated compared to when using the entire dataset. Contrary to this expectation, the novelty score for the SDM method seems lower than presented in the original paper (Table 2 of Hoogeboom et al.)''*
> >
> > We mention that Table 2 of Hoogeboom et al. *does not* present the novelty score (instead, it presents valid and unique). Indeed, their novelty score is presented in Table 5 of Hoogeboom et al., and the value is smaller than that presented in our paper. This inflation is as expected by Reviewer BGNw.
> >
> >
> > ## Response to Q5.
> > *``Also related, I’m unsure about how representative atom stability and molecular stability are as a metric. Especially in the context of this work where the bonds have to be estimated heuristically.''*
> >
> > Thanks for the comment. For fairness and completeness, we use the atom stability and molecular stability metrics following the EDM paper [2], although it is not our main focus.
> >
> > [1] LeCun et al., A Tutorial on Energy-Based Learning
> >
> > [2] Hoogeboom et al., Equivariant Diffusion for Molecule Generation in 3D
> >
> > [3] Zhao et al., EGSDE: Unpaired Image-to-Image Translation via Energy-Guided Stochastic Differential Equations

---

> > > ### Author Response · Authors · 2022-11-29
> > > **Sincerely looking forward to the further discussions**
> > >
> > > Dear reviewer BGNw,
> > >
> > > We are wondering if our response and revision have resolved your concerns. If our response has addressed your concerns, we would highly appreciate it if you could re-evaluate our work and consider raising the score.
> > >
> > > If you have any additional questions or suggestions, we would be happy to have further discussions.
> > >
> > > Best regards,
> > >
> > > The Authors

---

> > > ### Comment · Reviewer_BGNw · 2022-11-29
> > > **Thanks for clarification, but issue with MAE remain**
> > >
> > > Dear authors, thank you for your clarifications. Unfortunately, based on the comments to mine and other reviews, as well as the updated manuscript I do not feel comfortable increasing my score and recommending the paper for acceptance. The main reason for this is the use of the Mean-Absolute-Error as the main evaluation metric in the paper. The MAE is calculated using a second network purely trained to predict the property score for each generated molecule. There is no information given on the architecture of this network and the training setup used to train it. As a result of this, it is hard to determine if the predicted properties given by this network is an accurate representation of a structure's properties.
> > >
> > > While this issue can be partly alleviated by including more details about this secondary network in the paper, the dependency on a secondary network to derive the main evaluation metric is a more general concern.

---

> > > > ### Author Response · Authors · 2022-11-30
> > > > **Response to the further concern on MAE**
> > > >
> > > > Dear reviewer BGNw, thanks for your further comment.
> > > >
> > > > We argue that the predicted properties given by the secondary network is an accurate representation of a structure's properties, since the test loss of this network is very small (see L-bound in Table 1).
> > > >
> > > > Besides, when generating molecules with target structures, our method achieves a better similarity score than baselines, which does not depend on a secondary network. This is sufficient to demonstrate the effectiveness of our method.
> > > >
> > > >
> > > > We will present full details on the architecture and the training setup of the secondary network for MAE in the final version. Below are these details: we use the EGNN [4] as the secondary network following EDM, which has achieved superior performance in the task of the molecular property prediction on QM9 dataset. We use the provided official code and the default setting to train the secondary network, which can reproduce the results in EDM. Specifically, the EGNN consists of 7 layers and the number of hidden nodes is 128. We train the  secondary network for 1,000 epoch with Adam optimizer, batch size 96 and weight decay $10^{-16}$. The learning rates for the Homo, Lumo and Gap properties are $5\times 10^{-4}$ and for the others are $10^{-3}$.
> > > >
> > > > [4] Satorras et al., E(n) equivariant graph neural networks

---

> > > > ### Author Response · Authors · 2022-12-07
> > > > **Additional experiments that calculate MAE without a secondary network**
> > > >
> > > > In order to address the reviewer's concern on calculating the MAE metric using a second network, we add experiments that use the **Gaussian software** to compute properties of generated molecules, and then calculate MAE between properties of generated molecules (computed by Gaussian software) and desired properties. Note that the Gaussian software is one of the most popular computational chemistry software package, which calculates properties according to theories of quantum chemistry, and does not depend on neural networks.
> > > >
> > > > As shown in Rebuttal Table 1, our EEGSDE still has better MAE, which is consistent with Table 1 in the paper. This additional results further verify the effectiveness of our EEGSDE.
> > > >
> > > > Currently we compute 30 molecules for each method due to the limited time and we will include more molecules (e.g. 100) in the final version. We choose the calculation pipeline of Gaussian software, such that it can reproduce properties of molecules in QM9 dataset. We will release this pipeline in the final version.
> > > >
> > > > **Rebuttal Table 1.**   The mean
> > > > absolute error (MAE) between properties of generated molecules computed by Gaussian software and desired properties.
> > > > | Method          |  MAE  | Method          | MAE  | Method          | MAE | Method          | MAE | Method          | MAE |
> > > > |-----------------|------|-----------------|------|-----------------|-----|-----------------|-----|-----------------|-----|
> > > > |        $\mu$ (D)       |       |      $\alpha$ (Bohr$^3$)      |      |       $\Delta \varepsilon$ (meV)      |     |       $\varepsilon_{HOMO}$ (meV)      |     |       $\varepsilon_{LUMO}$ (meV)       |     |
> > > > | Conditional EDM | 1.16  | Conditional EDM | 2.3  | Conditional EDM | 717 | Conditional EDM | 354 | Conditional EDM | 561 |
> > > > | EEGSDE(s=0.5)   | 1.09  | EEGSDE(s=0.5)   | 2.25 | EEGSDE(s=0.5)   | 616 | EEGSDE(s=0.1)   | 350 | EEGSDE(s=0.5)   | 456 |
> > > > | EEGSDE(s=1)     | 0.97  | EEGSDE(s=1)     | 2.07 | EEGSDE(s=1)     | 563 | EEGSDE(s=0.5)   | 345 | EEGSDE(s=1)     | 384 |
> > > > | EEGSDE(s=2)     | **0.80**   | EEGSDE(s=3)     | **1.77**  | EEGSDE(s=3)     | **537**  | EEGSDE(s=1.0)   | **255**  | EEGSDE(s=3)     | **360**  |

---

> > > > > ### Comment · Reviewer_BGNw · 2022-12-09
> > > > > **Review updated**
> > > > >
> > > > > Thank you for this extended experiment. While I still worry about the conceptual significance of the improvement over EDM, these results show that there is a clear (statistical) significant improvement. I would be interested in getting the author's perspective on how conceptually significant this improvement is (eg. what is the distribution of the properties in the training set? Do the generated molecules fall within this range?), but this is not a requirement for me to accept the paper anymore. I will update my review.

---

> > > > > > ### Author Response · Authors · 2022-12-12
> > > > > > **Response to the new suggestion**
> > > > > >
> > > > > > Thanks for the new suggestion. We plot the distribution of the properties in the training set, as well as the distribution of the properties of molecules generated by our method (calculated by the Gaussian software). We find the two distributions match well. We will add these results in the final version.

---

### Author Response · Authors · 2022-12-07
**Additional experiments that calculate MAE using the Gaussian software**

Dear reviewers,

Inspired by the comment of Reviewer BGNw, we add experiments that use the **Gaussian software** to compute properties of generated molecules, and then calculate MAE between properties of generated molecules (computed by Gaussian software) and desired properties. Note that the Gaussian software is one of the most popular computational chemistry software package, which calculates properties according to theories of quantum chemistry, and does not depend on neural networks.

As shown in Rebuttal Table 1, our EEGSDE still has better MAE, which is consistent with Table 1 in the paper. This additional results further verify the effectiveness of our EEGSDE.

Currently we compute 30 molecules for each method due to the limited time and we will include more molecules (e.g. 100) in the final version. We choose the calculation pipeline of Gaussian software, such that it can reproduce properties of molecules in QM9 dataset. We will release this pipeline in the final version.


**Rebuttal Table 1.**   The mean
absolute error (MAE) between properties of generated molecules computed by Gaussian software and desired properties.
| Method          |  MAE  | Method          | MAE  | Method          | MAE | Method          | MAE | Method          | MAE |
|-----------------|------|-----------------|------|-----------------|-----|-----------------|-----|-----------------|-----|
|        $\mu$ (D)       |       |      $\alpha$ (Bohr$^3$)      |      |       $\Delta \varepsilon$ (meV)      |     |       $\varepsilon_{HOMO}$ (meV)      |     |       $\varepsilon_{LUMO}$ (meV)       |     |
| Conditional EDM | 1.16  | Conditional EDM | 2.3  | Conditional EDM | 717 | Conditional EDM | 354 | Conditional EDM | 561 |
| EEGSDE(s=0.5)   | 1.09  | EEGSDE(s=0.5)   | 2.25 | EEGSDE(s=0.5)   | 616 | EEGSDE(s=0.1)   | 350 | EEGSDE(s=0.5)   | 456 |
| EEGSDE(s=1)     | 0.97  | EEGSDE(s=1)     | 2.07 | EEGSDE(s=1)     | 563 | EEGSDE(s=0.5)   | 345 | EEGSDE(s=1)     | 384 |
| EEGSDE(s=2)     | **0.80**   | EEGSDE(s=3)     | **1.77**  | EEGSDE(s=3)     | **537**  | EEGSDE(s=1.0)   | **255**  | EEGSDE(s=3)     | **360**  |

---

### Author Response · Authors · 2023-04-13
**The official Code for EEGSDE**

The code is available at https://github.com/gracezhao1997/EEGSDE.

---

### Decision · Program_Chairs · 2023-01-20

**Decision:**

Accept: poster

**Justification For Why Not Higher Score:**

The methodological innovation is a bit limited

**Justification For Why Not Lower Score:**

The paper is very well written and gives a self contained introduction to equivariant diffusion models for molecule generation, and the paper demonstrates that energy guidance gives significantly better results on this task.

**Metareview: Summary, Strengths And Weaknesses:**

This paper presents a way to guide equivariant diffusion models for molecules using an energy function. The paper is very well written, and shows significant improvements. The main concern voiced by reviewers is that adding an energy function for guidance of diffusion models is a well known method, which just hasn't been applied to equivariant diffusion models for molecule generation. As such, the methodological innovation is a bit limited, but the paper does show that this approach yields significant gains. Together with the educational value of the paper, this makes it well worth publishing in my view.

One further concern that was voiced was that the MAE is evaluated via a secondary classifier, but this issue has been addressed by also evaluation using the Gaussian software.

After discussions, three of the four reviewers now agree that the paper should be accepted, and the last one has reported being sick and thus has not been able to consider the latest discussions.

**Note From Pc:**

if the above contains the word "oral" or "spotlight" please see: "oral" presentation means -> notable-top-5% and "spotlight" means -> notable-top-25%. As stated in our emails, we are disassociating presentation type from AC recommendations

**Summary Of Ac-Reviewer Meeting:**

We had a meeting, though due to a last-minute cancellation (sickness), and one reviewer being unavailable (house viewing), only pro-accept reviewers could make it (two). After I shared the meeting notes, one negative reviewer (who did not attend the meeting) changed his score from 5 to 6. I'm fairly confident we should accept the paper, despite one remaining 5 rating.